# ANYTOUCH: LEARNING UNIFIED STATIC-DYNAMIC REPRESENTATION ACROSS MULTIPLE VISUO-TACTILE SENSORS

**Ruoxuan Feng**[1]    **Jiangyu Hu**[2,3]    **Wenke Xia**[1]    **Tianci Gao**[1]    **Ao Shen**[1]    **Yuhao Sun**[3]
**Bin Fang**[3*]   **Di Hu**[1*]
[1]Renmin University of China    [2]Wuhan University of Science and Technology
[3]Beijing University of Posts and Telecommunications

## ABSTRACT

Visuo-tactile sensors aim to emulate human tactile perception, enabling robots to precisely understand and manipulate objects. Over time, numerous meticulously designed visuo-tactile sensors have been integrated into robotic systems, aiding in completing various tasks. However, the distinct data characteristics of these low-standardized visuo-tactile sensors hinder the establishment of a powerful tactile perception system. We consider that the key to addressing this issue lies in learning unified multi-sensor representations, thereby integrating the sensors and promoting tactile knowledge transfer between them. To achieve unified representation of this nature, we introduce **TacQuad**, an aligned multi-modal multi-sensor tactile dataset from four different visuo-tactile sensors, which enables the explicit integration of various sensors. Recognizing that humans perceive the physical environment by acquiring diverse tactile information such as texture and pressure changes, we further propose to learn unified multi-sensor representations from both static and dynamic perspectives. By integrating tactile images and videos, we present **AnyTouch**, a unified static-dynamic multi-sensor representation learning framework with a multi-level structure, aimed at both enhancing comprehensive perceptual abilities and enabling effective cross-sensor transfer. This multi-level architecture captures pixel-level details from tactile data via masked modeling and enhances perception and transferability by learning semantic-level sensor-agnostic features through multi-modal alignment and cross-sensor matching. We provide a comprehensive analysis of multi-sensor transferability, and validate our method on various offline datasets and in the real-world pouring task. Experimental results show that our method outperforms existing methods, exhibits outstanding static and dynamic perception capabilities across various sensors. The code, TacQuad dataset and AnyTouch model are fully available at gewu-lab.github.io/AnyTouch/.

## 1 INTRODUCTION

Tactile perception is an important sense through which humans perceive the physical world. For many years, researchers have been working to endow robots with human-like tactile perception abilities through diverse tactile sensors (Liu et al., 2022; Maiolino et al., 2013; Yuan et al., 2017). Among them, with high resolution comparable to human touch, various types of visuo-tactile sensors have garnered widespread attention (Yuan et al., 2017; Donlon et al., 2018; Lambeta et al., 2020). Many studies have attempted to use robots equipped with visuo-tactile sensors to perform manipulation tasks such as grasping (Xu et al., 2024) and inserting (Li et al., 2014).

However, due to the low standardization of visuo-tactile sensors, different sensors may exhibit discrepancies in perceiving the same tactile information. This variability poses challenges to building precise robotic tactile systems, as sensor-specific data collection (Yang et al., 2022; Gao et al., 2023) and model training limit the data scale and diversity for the model of a single sensor and lead to suboptimal perception capabilities. To address this issue, some initial efforts have explored us-

---

*Equal corresponding authors.

ing multi-sensor data collaboratively to enhance cross-sensor knowledge transferability (Yang et al., 2024; Zhao et al., 2024). Nevertheless, the lack of aligned multi-sensor data has hindered these attempts from effectively integrating disparate sensors and constructing a unified representation space. Considering this data issue, Rodriguez et al. (2024) collected a dual-sensor paired dataset to enable cross-sensor generation. However, their focus on specific manipulation tasks limited the variety of sensors and collected objects. Moreover, they overlooked the potential benefits of paired multi-modal data for enhancing sensor transferability and achieving comprehensive tactile perception.

To enhance support for multi-sensor integration, we collect **TacQuad**, an aligned multi-modal multi-sensor tactile dataset containing 72,606 contact frames, using four different visuo-tactile sensors. We select these representative sensors from publicly available sensors, self-made sensors, and force field sensors to ensure diversity. To balance the trade-off between the cost of data collection and the accuracy of pairing, we collect fine-grained spatio-temporal aligned data on a calibration platform, while larger-scale coarse-grained spatial aligned data is acquired through the handheld collection, as shown in Figure 1. Additionally, we capture the objects being touched using a camera and annotate tactile attribute descriptions for each collection, forming a comprehensive touch-vision-language dataset. This dataset utilizes paired multi-modal data as a bridge to mitigate the impact of sensor variability on the understanding of tactile semantic features, and enables the explicit integration of sensors into a unified multi-sensor space for effective knowledge transfer between sensors.

Building on this solid foundation, we further revisit the challenge of learning unified multi-sensor representations: How can we obtain unified multi-sensor representations adaptable to a wide array of tasks? We recognize that the human tactile perception is a combination of static and dynamic processes, as humans derive comprehensive tactile perception from multiple types of information such as texture, sliding, and pressure changes. Drawing on this insight, we propose *learning unified representations from both static and dynamic perspectives* to accommodate a range of tasks.

To obtain multi-sensor representations of this nature, we introduce **AnyTouch**, a unified static-dynamic multi-sensor tactile representation learning framework. This framework integrates the input forms of tactile images and videos, collectively utilizing them to reinforce the model's abilities to perceive both static properties and dynamic changes. Moreover, we design a multi-level architecture to comprehensively strengthen the model's capabilities for capturing pixel-level tactile details and semantic-level sensor-agnostic features. Specifically, we utilize masked modeling (He et al., 2022; Tong et al., 2022) to maximize the use of multi-sensor data for learning fine-grained, pixel-level details. Subsequently, we conduct multi-modal aligning and a novel cross-sensor matching task to understand semantic-level tactile properties of objects across different sensors and extract sensor-agnostic features. We aim for the multi-sensor representations to share a common space and cluster by the tactile information of the object they represent, thereby reducing the gap between sensors. To further promote knowledge transfer across multiple sensors, we propose randomly replacing the sensor-specific tokens (Yang et al., 2024) with universal sensor tokens during training. This strategy ensures the model maintains its ability to process and perceive tactile data across various sensors, while also providing knowledge from all seen sensors for generalization to unseen sensors.

We conduct both quantitative and qualitative experiments to analyze the transferability of multi-sensor data and assess the impact of our framework on the multi-sensor representation space. Building on this, we comprehensively evaluate the static and dynamic tactile perception capabilities of AnyTouch across various tactile datasets and through a real-world experiment: fine-grained pouring. The experimental results demonstrate the static and dynamic perception abilities and cross-sensor transferability of AnyTouch. We hope the approach of learning unified multi-sensor representations from both static and dynamic perspectives will establish a standardized learning paradigm for visuo-tactile perception and further inspire research in multi-sensor representation learning.

## 2 RELATED WORK

**Multi-Source Learning.** Learning from multi-source data with greater data scale and diversity is expected to enhance the model's performance and generalization ability, but faces challenges in integrating the data representation spaces. Researchers have found that multi-source models often struggle to capture a unified representation suffering from the discrepancies between data sources (Glorot et al., 2011; Zhao et al., 2019). Contrastive learning (Wang et al., 2022b; Zhao et al., 2023) is proposed to learn language-agnostic representations for multi-source language training, while cycle consistency loss (Zhu et al., 2017; Kim et al., 2022) aligns the target domain distributions in multi-

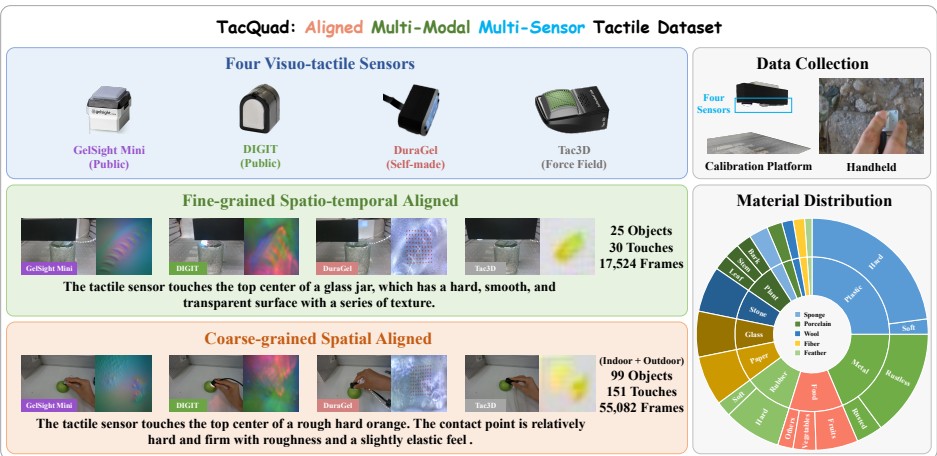

Figure 1: **TacQuad: an aligned multi-modal multi-sensor tactile dataset from four visuo-tactile sensors.** We select GelSight Mini (Inc.) and DIGIT (Lambeta et al., 2020) from publicly available sensors, DuraGel (Zhang et al., 2024) from self-made sensors, and Tac3D (Zhang et al., 2022) from force field sensors for diversity. There is a noticeable gap between the data from these sensors. We use the four sensors to touch the same position on the same object to obtain aligned data. To maximize aligned data collection, we use two methods to gather subsets with different alignment accuracy. We collect fine-grained spatio-temporal aligned data on a calibration platform, while larger-scale coarse-grained spatial aligned data is acquired through handheld collection.

source image generation. Similarly, to integrate multi-source tactile data from different sensors, techniques such as multi-sensor joint training (Zhao et al., 2024; Higuera et al., 2024; Gupta et al., 2025), multi-modal alignment (Yang et al., 2024), and cross-sensor generation (Rodriguez et al., 2024) have emerged. However, these methods overlook the benefits of jointly utilizing multi-modal data and aligned multi-sensor data to bridge the sensor gap. In this work, we collect an aligned multi-modal multi-sensor dataset and propose learning unified multi-sensor representations.

**Visuo-tactile Perception.** Visuo-tactile sensors have garnered widespread attention due to their high resolution (Yuan et al., 2017; Donlon et al., 2018; Lambeta et al., 2020; Zhang et al., 2024; 2025). Nowadays, many works utilize visuo-tactile sensors to capture contact deformations, enabling the completion of dexterous manipulation such as dense packing (Li et al., 2022), grasping (Xu et al., 2024), and insertion (Li et al., 2014). Visuo-tactile sensors can also collaborate with other sensors to assist the robot in performing more complex manipulation tasks. For instance, Feng et al. (2024) dynamically integrates visuo-tactile signals with visual and audio data to collaboratively accomplish fine-grained manipulation tasks, such as pouring and peg insertion with keyway. In addition to these tasks requiring dynamic tactile perception, visuo-tactile sensors are also used in static tasks such as material classification (Yang et al., 2022) and shape reconstruction (Gao et al., 2022b). However, due to the low standardization of visual-tactile sensors, these methods fail to leverage larger and more diverse data from other sensors and lack sensor transferability. In this work, we propose learning a unified multi-sensor representation from both static and dynamic perspectives.

**Representation Learning.** Representation learning has achieved remarkable success in improving model generalization in various fields. Techniques like BERT (Devlin, 2018) and masked autoencoder (MAE) (He et al., 2022) have enhanced the model's performance across various downstream applications. With the rise of multi-modal learning, representation learning has expanded its impact across fields. Vision-language pre-training (Radford et al., 2021) has seen tremendous success, and more modalities, including audio (Guzhov et al., 2022), touch (Yang et al., 2024), and 3D point clouds (Xue et al., 2023), are being integrated. Among them, tactile information from visuo-tactile sensors can be expressed as images, allowing vision-related techniques to make strides in touch. Applying MAE (Cao et al., 2023; Higuera et al., 2024) or multi-modal aligning (Yang et al., 2024; Cheng et al., 2024) has enhanced tactile model capabilities. However, these efforts have not explored how to obtain a unified visuo-tactile representation suitable for various tasks. Our research addresses this challenge from both static and dynamic perspectives, enhancing cross-sensor transferability across various tasks through semantic-level multi-modal aligning and cross-sensor matching.

## 3 ALIGNED MULTI-MODAL MULTI-SENSOR TACTILE DATASET

The low standardization of visuo-tactile sensors and the gap between multi-sensor data have resulted in insufficient data for individual sensors and the poor cross-sensor transferability of tactile models. Rodriguez et al. (2024) has made an initial attempt to address it by collecting a dataset with 32,256 pairs of tactile images from two sensors with a limited variety of objects for specific manipulation tasks. It does not consider tactile properties such as material and hardness, and overlooks the potential to enhance cross-sensor transfer capabilities through multi-modal information.

In this work, we present a more comprehensive solution to this problem by providing multi-sensor aligned data with text and images, explicitly enabling the model to learn semantic-level tactile attributes and sensor-agnostic features to form a unified multi-sensor representation space through data-driven approaches. We collect TacQuad, an aligned multi-modal multi-sensor tactile dataset with a greater variety of objects, larger data volume, and more types of sensors. To ensure sensor diversity, we select GelSight Mini (Inc.) and DIGIT (Lambeta et al., 2020) from publicly available sensors, DuraGel (Zhang et al., 2024) from self-made sensors, and Tac3D (Zhang et al., 2022) from force field sensors for data collection. The first three sensors are used to collect tactile images, while the Tac3D is used to capture deformation force fields. However, considering collecting fine-grained aligned tactile data is very costly, to collect data on a larger scale while ensuring as much data pairing as possible, we use both coarse and fine methods to collect aligned data:

**Fine-grained spatio-temporal aligned data**. We fix the four sensors side by side in a rectangular container and connect them to the movable end of a calibration platform. We use a program to control the four sensors to press the same position on the same object in sequence, sharing the same speed and depth. Consequently, we obtain a set of continuous spatio-temporally aligned contact frames. Due to the high requirement of precision, the process is time-consuming, thus we try our best to collect 30 sets of aligned data across 25 objects. This portion of the data contains a total of 17,524 contact frames, which can be used for fine-grained tasks such as cross-sensor generation. See more details for the fine-grained data collection in Sec. A.5 of the Appendix.

**Coarse-grained spatial aligned data**. We collect data in a handheld manner by sequentially pressing the same location on the same object with four sensors. While pressing, we introduce some twisting motions to the handheld sensors to better simulate the authentic dynamic touch experience of humans. This method allows us to obtain a larger amount of aligned data in a short time. Using this approach, we collect 151 sets of aligned data from 99 objects, including both indoor and outdoor scenes. This portion of the data contains a total of 55,082 contact frames.

Each tactile frame in the dataset has a paired visual image and tactile attribute descriptions that generated using GPT-4o and manually corrected. We aim to bridge the gap between sensors and achieve a more comprehensive tactile perception by aligning with the multi-modal data. As a result, we obtain an aligned multi-sensor multi-modal tactile dataset, as shown in Figure 1.

## 4 METHOD

In this section, we introduce AnyTouch, a unified multi-sensor tactile representation learning framework from the perspectives of both static and dynamic perception, as shown in Figure 2. Concretely, we integrate the input format of tactile images and videos (Sec. 4.1) and focus on learning both fine-grained pixel-level details for refined tasks (Sec. 4.2) and semantic-level sensor-agnostic features for understanding properties (Sec. 4.3) and building unified space (Sec. 4.4) by a multi-level structure. We also propose universal sensor tokens for better knowledge transfer.

### 4.1 UNIFIED INPUT FORMAT FOR STATIC AND DYNAMIC TACTILE PERCEPTION

In daily life, human tactile perception includes both static and dynamic processes. A brief touch allows quick recognition of properties like material and texture, while tasks such as unlocking a lock require continuous dynamic perception. These two types of perception complement each other, enabling us to comprehensively understand the physical surroundings and engage in a variety of interactions. This inspires us to learn unified multi-sensor representation from the perspective of combining static and dynamic perception, using tactile images and videos respectively.

Given a static tactile image $I \in \mathbb{R}^{1 \times H \times W \times 3}$ and a dynamic tactile video clip $V \in \mathbb{R}^{F \times H \times W \times 3}$, we consider tactile images as single-frame static videos to unify tactile images and videos. Concretely, we replicate $I$ along the time axis for $F$ times, and use a unified 4-D tensor $X_T \in \mathbb{R}^{F \times H \times W \times 3}$

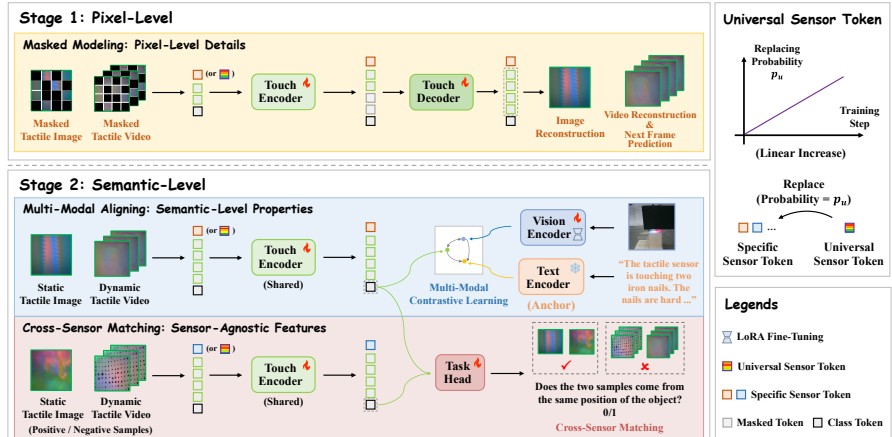

Figure 2: **Overview of AnyTouch.** Our framework integrates static tactile images and dynamic tactile videos, aiming to learn a unified multi-sensor representation suitable for various tasks. Through a multi-level architecture, we employ masked modeling to learn pixel-level tactile details, and use multi-modal aligning and cross-sensor matching to understand semantic-level sensor-agnostic tactile properties. We also use universal sensor tokens to integrate and transfer sensor information.

to represent both $I$ and $V$ as Girdhar et al. (2022; 2023), where $F$ is the number of frames and $H, W$ denote the shape of images. We then process $X_T \in \mathbb{R}^{F \times H \times W \times 3}$ into spatio-temporal tokens $z \in \mathbb{R}^{N \times d}$ through a shared patch projection layer, where $N$ is the length of tokens and $d$ represents the feature dimension. By unifying the processing of images and videos in this manner, our approach integrates tactile images and video input, enhancing the model's ability to comprehend both static and dynamic information, and endows the model with the potential to accomplish various tasks.

## 4.2 MASKED MODELING: LEARNING PIXEL-LEVEL DETAILS

Visuo-tactile images are fine-grained data with pixel-level details of subtle tactile deformations and continuous changes during dynamic processes, especially for refined perception tasks. To enhance the fine-grained perception capabilities of the tactile representation model, we employ the masked autoencoder technique (He et al., 2022; Tong et al., 2022), compelling the model to capture pixel-level details across multiple sensors. Concretely, we randomly mask the tokens of both tactile images and videos with a masking ratio $\rho$, and build a decoder to obtain the reconstructed static images $\hat{I}$ and dynamic videos $\hat{V}$. The corresponding loss function $\mathcal{L}_{rec}^{S}$ and $\mathcal{L}_{rec}^{D}$ are *mean squared error* (MSE) loss between the original masked tokens and reconstructed ones in the pixel space:

$$\mathcal{L}_{rec}^{S} = \frac{1}{|\Omega_M|} \sum_{p \in \Omega_M} |\hat{I}(p) - I(p)|^2, \;\; \mathcal{L}_{rec}^{D} = \frac{1}{F|\Omega_M|} \sum_{f}^{F} \sum_{p \in \Omega_M} |\hat{V}_f(p) - V_f(p)|^2, \tag{1}$$

where $p$ is the token index, $\Omega_M$ is the set of masked tokens and $V_f$ is the $f$-th frame in the video $V$. We use masked modeling to learn fine-grained tactile deformation features at the pixel level, as well as the temporal dynamics of tactile changes.

To further enhance the model's understanding of continuous deformation changes, we introduce an additional task of predicting the next frame $V_{F+1}$ while reconstructing the dynamic video $V$. The loss function $\mathcal{L}_{pred}^{D}$ is MSE loss between the original frame $V_{F+1}$ and the predicted frame $\hat{V}_{F+1}$:

$$\mathcal{L}_{pred}^{D} = \frac{1}{N} \sum_{p}^{N} |\hat{V}_{F+1}(p) - V_{F+1}(p)|^2. \tag{2}$$

## 4.3 MULTI-MODAL ALIGNING: UNDERSTANDING SEMANTIC-LEVEL PROPERTIES

After obtaining tactile representations with fine-grained perceptual details via masked modeling, we aim to further understand semantic-level tactile properties and use paired multi-modal data as a bridge to narrow the gap between sensors. Therefore, we propose using multi-modal aligning, which binds data from various sensors with paired modalities for a more comprehensive semantic-level perception and reduce perceptual differences between sensors. However, differences in data collection scenarios across various datasets (*e.g.*, simulation vs. reality) make simple vision-tactile

Figure 3: **Comparison with existing multi-modal aligning methods.** Our method not only uses multi-modal data to bridge the gap between sensors, but also **explicitly** clusters representations of the same position on the same object from different sensors together.

alignment less effective in bridging sensor gaps. Therefore, we select the text modality, which consistently describes tactile attributes across datasets, as an anchor to align touch, vision, and text modalities. Since tri-modal tactile datasets are rare, with most containing only vision-touch pairs, we explore two strategies: automatically expanding the amount of text modality pairings and designing aligning methods that are compatible with missing modalities. We first select representative datasets for each sensor and then use GPT-4o to generate or expand the text modality within these datasets. Through this method, we create new text pairs for 1.4 million samples across the four datasets.

Based on these extensive tactile datasets, we develop a modality-missing-aware touch-vision-language contrastive learning method to leverage the paired data between touch and other modalities for alignment. We maximize the use of paired data by selecting the largest subset for each modality combination within the batch for multi-modal aligning. Considering a pair of uni-modal representations $(x_T, x_V, x_L)$ derived from uni-modal encoders, where $x_T \in \mathbb{R}^d$ is the touch representation, $x_V \in \mathbb{R}^d \cup \varnothing$ is the vision representation and $x_L \in \mathbb{R}^d \cup \varnothing$ is the text representation. We then perform multi-modal alignment (Radford et al., 2021) within the batch as (taking vision-language aligning with missing modalities as an example):

$$\mathcal{L}_{V \to L} = -\frac{1}{|\Omega_V \cap \Omega_L|} \sum_{i \in \Omega_V \cap \Omega_L} \log \frac{\exp(x_{V,i}^\top \cdot x_{L,i}/\tau)}{\sum_{j \in \Omega_v \cap \Omega_L} \exp(x_{V,i}^\top \cdot x_{L,j}/\tau)}, \quad (3)$$

where $B$ is the batchsize, $\Omega_V, \Omega_L$ are sets of indices for the samples containing vision and text, and $\tau$ is the scalar temperature. The computation of similar $\mathcal{L}_{T \to V}$ and $\mathcal{L}_{T \to L}$ is shown in Sec. A.2 of the Appendix. This approach maximizes the use of paired data with missing modalities by aligning the sample intersections between modalities. The computation of $\mathcal{L}_{V \to T}$, $\mathcal{L}_{L \to T}$ and $\mathcal{L}_{L \to V}$ is similar but in the opposite direction. We then obtain the joint aligning loss as:

$$\mathcal{L}_{align} = \frac{\alpha_{TV}}{2}(\mathcal{L}_{T \to V} + \mathcal{L}_{V \to T}) + \frac{\alpha_{TL}}{2}(\mathcal{L}_{T \to L} + \mathcal{L}_{L \to T}) + \frac{\alpha_{VL}}{2}(\mathcal{L}_{V \to L} + \mathcal{L}_{L \to V}), \quad (4)$$

where $\alpha_{TV}$, $\alpha_{TL}$ and $\alpha_{VL}$ are hyper-parameters to control the alignment strength.

### 4.4 CROSS-SENSOR MATCHING: EXTRACTING SENSOR-AGNOSTIC FEATURES

To fully utilize multi-sensor aligned data and build unified space by clustering multi-sensor tactile representations of the same object, we introduce a novel cross-sensor matching task. In this task, the model needs to determine whether two tactile images or videos are collected from the same position on the same object. We aim to cluster representations of the same tactile information from different sensors while performing multi-modal aligning, thereby enhancing the learning of sensor-agnostic features and forming a unified multi-sensor representation space, as shown in Figure 3.

We treat data collected from the same object and position by two different sensors as a positive pair, and data from different objects or positions as a negative pair. The model is trained to distinguish between positive and negative pairs. For each image and video sample $X_T$ in our TacQuad, we randomly select one sample from the same object at the same location captured by another sensor as the positive sample $X_T^+$, and choose another sample from any dataset of any other object or location as a negative sample $X_T^-$. We element-wisely multiply the touch representation $x_T$ with $x_T^+$ and $x_T^-$, and then input each result into an MLP to compute the matching scores $m^+$ and $m^-$:

$$m^+ = MLP(x_T \cdot x_T^+), \ m^- = MLP(x_T \cdot x_T^-), \quad (5)$$

where $x_T$, $x_T^+$ and $x_T^-$ are the representations of $X_T$, $X_T^+$ and $X_T^-$. The loss function $\mathcal{L}_{match}$ is a Binary Cross Entropy Loss similar to Lin et al. (2020):

$$\mathcal{L}_{match} = -(y^+ \log(m^+) + (1 - y^+) \log(1 - m^-)) - (y^- \log(m^-) + (1 - y^-) \log(1 - m^-)). \quad (6)$$

This task requires the model to distinguish features with the same semantics from different sensors, thus explicitly clustering representations with the same object information form a unified multi-sensor representation space. As shown in Figure 3, AnyTouch, incorporating this task, differs from existing multi-modal aligning efforts. The construction of this unified multi-sensor representation space can explicitly reduce the gap between sensors and aid in generalizing to unseen sensors.

As both this task and multi-modal aligning focus on semantic-level features, we combine them as the second stage, with masked modeling as the first stage. This multi-level training approach allows us to develop unified multi-sensor representations adaptable to tasks of varying granularities.

## 4.5 Universal Sensor Token

In addition to building a multi-sensor representation space, we aim to extract and store information related to each sensor to aid the understanding of input data. More importantly, we want to integrate and effectively transfer this information when generalizing to new sensors. Using sensor-specific tokens is a method for extracting sensor-specific information, but this approach cannot fully transfer information from all seen sensors when generalizing to new sensors (Yang et al., 2024).

Therefore, we propose using universal sensor tokens to integrate and store information related to various sensors, thereby maximizing the utilization of multi-sensor data when generalizing to new sensors. During training, we randomly replace the sensor-specific tokens with the universal sensor tokens, expecting them to aid in understanding input data from various sensors. Specifically, we introduce a set of learnable sensor tokens $\{s_k\}_{k=1}^K \cup s_u$, where $K$ is the number of sensor types, $s_k \in \mathbb{R}^{L \times d}$ are the sensor-specific tokens for the $k$-th sensor, $s_u \in \mathbb{R}^{L \times d}$ are universal sensor tokens and $L$ is the number of sensor tokens for each sensor. When inputting the tactile token sequence $z$ from the $k$-th sensor into the encoder $\Phi_{enc}$ to obtain its representation $x_T$, we randomly select one from $s_k$ and $s_u$ to concatenate with $z$, as follows:

$$s = i \cdot s_u + (1 - i) \cdot s_k, \; i \sim B(p_u),$$
$$x_T = \Phi_{enc}(z, s), \tag{7}$$

where $p_u$ is the probability of using universal sensor tokens $s_u$. During inference, we consistently use universal sensor tokens for data from new sensors.

## 4.6 Training Paradigm

Our framework has a multi-level structure, with the training of two stages conducted sequentially. In the first stage, we simultaneously perform the reconstruction of static tactile images and dynamic tactile videos, as well as the unique next frame prediction task for tactile videos. The loss for the first stage $\mathcal{L}_{stage1}$ is as follows:

$$\mathcal{L}_{stage1} = \mathcal{L}_{rec}^S + \mathcal{L}_{rec}^D + \mathcal{L}_{pred}^D. \tag{8}$$

In the second stage, we continue to use both tactile images and videos, and simultaneously perform multi-modal aligning and cross-sensor matching tasks. Hence, the loss function for the second stage is the sum of the losses from these two tasks:

$$\mathcal{L}_{stage2} = \mathcal{L}_{align} + \lambda \mathcal{L}_{match}, \tag{9}$$

where $\lambda$ is a hyper-parameter controlling the weight of cross-sensor matching task.

## 5 Experiments

In this section, we explore the answers to the following questions through quantitative and qualitative experiments: (**Q1**) How much benefit does the data of each sensor provide when it is integrated? (**Q2**) What does the unified multi-sensor representation space constructed by AnyTouch look like? (**Q3**) Is the unified multi-sensor representation more advantageous in various static and dynamic perception tasks? We analyze **Q1** in Section 5.2, expolore **Q2** in Section 5.3, and answer **Q3** through comparisons with existing methods in Sections 5.4, 5.5 and 5.6.

## 5.1 Dataset and Baselines

We use 9 different tactile datasets for training, including: Touch and Go (TAG) (Yang et al., 2022), VisGel (Li et al., 2019), Cloth (Yuan et al., 2018), ObjectFolder Real (OF Real) (Gao et al., 2023) ,

Table 1: The impact of adding data from multiple sensors on the seen dataset (TAG), the unseen dataset from seen sensors (Feel), and the unseen dataset from unseen sensors (OF 1.0 and OF 2.0).

| Tactile Training Data | Data Volume | TAG Material | Feel Grasp | OF 1.0 Material | OF 2.0 Material |
|---|---|---|---|---|---|
| No Tactile Pre-training (CLIP) | / | 52.96 | 72.37 | 41.00 | 73.16 |
| TAG, VisGel, Cloth | 996k | **83.55** (↑30.59) | 79.12 (↑6.75) | 46.12 (↑5.12) | 75.10 (↑1.94) |
| TAG, VisGel, Cloth, OF Real | 2161k | 79.67 (↓3.88) | 79.28 (↑0.16) | 47.55 (↑1.43) | 75.53 (↑0.43) |
| TAG, VisGel, Cloth, OF Real, TVL, SSVTP, YCB-Slide | 2388k | 79.61 (↓0.06) | 79.10 (↓0.18) | 48.00 (↑0.45) | 75.57 (↑0.04) |
| TAG, VisGel, Cloth, OF Real, TVL, SSVTP, YCB-Slide, Octopi | 2427k | 79.70 (↑0.09) | **79.40** (↑0.30) | **48.75** (↑0.75) | **75.66** (↑0.09) |

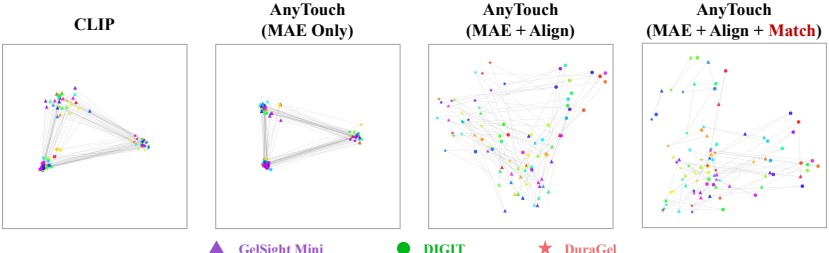

Figure 4: **The impact of components in AnyTouch on the multi-sensor representation space.** We use t-SNE to visualize the representations on the unused fine-grained subset of TacQuad, starting with CLIP and sequentially incorporating the modules. Each color represents a single touch, and samples from three sensors that touch the same position are connected by dashed lines.

TVL (Fu et al., 2024), YCB-Slide (Suresh et al., 2023) and SSVTP (Kerr et al., 2022), Octopi (Yu et al., 2024) and the coarse-grained subset of our TacQuad. We leverage the continual frames available in these datasets for dynamic perception. We also select four datasets: TAG, Feel (Calandra et al., 2017), ObjectFolder 1.0 (Gao et al., 2022a), ObjectFolder 2.0 (Gao et al., 2022b) as the downstream datasets. We compare AnyTouch with several single-sensor models: VIT-LENS-2 (Lei et al., 2024), TLV-Link (Cheng et al., 2024), and Omnibind (Lyu et al., 2024). We also compare our method with the multi-sensor model UniTouch (Yang et al., 2024). We use the largest subset of all data that meets the requirements of UniTouch and TLV-Link to train them, remarked as UniTouch† and TLV-Link†. For the real-world dynamic perception task, we compare our method with a multi-sensor model T3 (Zhao et al., 2024), which is trained on 3M data, more than AnyTouch. The detailed dataset and baseline introduction are provided in Appendix A.1, A.3 and A.4.

## 5.2 SENSOR TRANSFERABILITY ANALYSIS (Q1)

Since we have integrated data from multiple sensors into unified representations, we aim to investigate the contributes of the knowledge transferred from each sensor's data to downstream tasks. Therefore, we incorporate data from GelSight, GelSlim (Donlon et al., 2018), DIGIT and GelSight Mini into the training of AnyTouch to obtain four different models, and compare them across four downstream tasks. As shown in Table 1, training with only GelSight data significantly improves performance on downstream datasets for all sensors, compared to the CLIP model which has not encountered any tactile data. This indicates that tactile representation pre-training is crucial and transferable across new sensors. After sequentially integrating data from GelSlim, DIGIT, and GelSight Mini into the training, we observe performance improvements across the three unseen datasets, with greater enhancements for unseen sensors than seen sensors. This suggests that the knowledge from the data of GelSlim, DIGIT, and GelSight Mini can transfer to the GelSight and other sensors.

However, we also observe two interesting phenomena: (1) In the material classification task of the seen dataset TAG, the model trained solely on GelSight data performs the best, while incorporating data from more sensors leads to a performance drop. This is because TAG is included in the pre-training data and integrating more data reduces the proportion of TAG data in pre-training. This aligns with the CLIP paper's finding that greater overlap between the downstream task dataset and the pre-training dataset may improve performance. (2) Although the integrated data volume from

Table 2: Evaluation of static perception capabilities on the seen sensor (GelSight). *Note that Feel is seen for the corresponding UniTouch and AnyTouch models. ‡Note that original TLV-Link uses frames after grasping which are much easier in Feel, whereas other models use frames during grasping. AnyTouch achieves a result of 99.0 using frames after grasping.

| Method | Tactile Training Data | Touch and Go | | | Feel |
| | | Material | Roughness | Hardness | Grasp |
|---|---|---|---|---|---|
| CLIP | / | 52.96 | 84.09 | 88.34 | 72.37 |
| VIT-LENS-2 | TAG | 63.0 | 85.1 | 92.0 | - |
| TLV-Link | Touch100k | 67.2 | 84.7 | 91.3 | 94.5‡ |
| OmniBind | TAG | 67.45 | - | - | - |
| UniTouch | TAG, Feel*, YCB, OF 2.0 | 61.3 | - | - | 82.3 |
| TLV-Link† | TAG, TVL, SSVTP, OF Real, TacQuad | 74.12 | 85.94 | 94.18 | 76.97 |
| **AnyTouch** | TAG, Feel*, YCB, OF 2.0 | **82.74** | 86.01 | 94.24 | **87.17** |
| **AnyTouch** | TAG, VisGel, Cloth, TVL, SSVTP, YCB-Slide, OF Real , Octopi, TacQuad | 80.82 | **86.74** | **94.68** | 80.53 |

DIGIT is larger, the benefits are less compared to incorporating data from GelSight Mini. This may suggest that the images from DIGIT differ more from the images of GelSight and other sensors than the images from GelSight Mini do due to the hardware difference.

## 5.3 MULTI-SENSOR REPRESENTATION SPACE (**Q2**)

To verify whether AnyTouch clusters the representations with the same tactile information from different sensors together as expected, we use t-SNE (Van der Maaten & Hinton, 2008) to visualize the tactile representations. We extract one aligned contact frame from each sensor for the 30 touches in the unused fine-grained subset of TacQuad. We input these samples into the CLIP model and the AnyTouch model which gradually incorporates masked modeling, multi-modal aligning, and cross-sensor matching, and visualize their representations in Figure 4. Due to the lack of exposure to tactile images, CLIP struggles to distinguish the same tactile information from different sensors, instead clustering samples by sensor. After introducing masked modeling, the representations become more centralized within each sensor, as this method focuses on pixel-level tactile features, which are sensor-dependent. However, this is not ideal for cross-sensor generalization, as we want multi-sensor tactile representations to cluster based on the object's tactile information they represent, minimizing sensor gaps. After incorporating multi-modal aligning, the multi-sensor tactile representations begin to blend and cluster by the objects they represent. This indicates that cross-sensor generalization is beginning to emerge, but there is still a distinct tendency for sensor-specific clustering. With our cross-sensor matching task, the representations from different sensors fully mix in a shared multi-sensor space, clearly clustering by the object's tactile information. This indicates that our model possesses the ability to extract sensor-agnostic features, enabling generalization to unseen sensors.

## 5.4 STATIC PERCEPTION ON SEEN SENSORS (**Q3**)

To validate the benefit of unified multi-sensor representations in transferring knowledge from multiple sensor data to seen sensors, we compared it to baselines on the seen dataset TAG and the unseen dataset Feel from the GelSight sensor. Since UniTouch uses different training data, we also train an AnyTouch model with the same data as UniTouch to ensure a fair comparison. As shown in Table 2, the TLV-Link† trained with multi-sensor data outperforms all single-sensor models and the original TLV-Link in all three tasks of TAG. The AnyTouch trained with the same data as UniTouch, outperforms UniTouch in all four tasks. With the integration of dynamic perception and more multi-sensor data, AnyTouch trained on all data achieved the best results in hardness and roughness classification in TAG and comparable results to UniTouch in Feel, despite UniTouch having seen this data. These demonstrate the static perception capabilities of our framework on seen sensors. Notably, the original TLV-Link uses frames after grasping, while other models use frames during grasping. AnyTouch achieves a result of 99.0 using frames after grasping. It is worth mentioning that the AnyTouch trained with less data outperforms the one trained with all data in TAG material classification, similar to Table 1, while exposure to more multi-sensor data enhances performance in hardness and roughness classification. This is because these binary hardness and roughness classification tasks

Table 3: Evaluation of static perception capabilities on unseen sensors (TACTO and Taxim) using linear probing. *Note that OF 2.0 is seen for the corresponding UniTouch and AnyTouch models.

| Method | Tactile Training Data | ObjectFolder 1.0 Material | ObjectFolder 2.0 Material |
|---|---|---|---|
| CLIP | / | 41.00 | 73.16 |
| UniTouch | TAG, Feel, YCB-Slide, OF 2.0* | 41.3 | 85.4 |
| UniTouch† | TAG, VisGel, TVL SSVTP, OF Real , TacQuad | 47.25 | 75.29 |
| **AnyTouch** | TAG, Feel, YCB-Slide, OF 2.0* | 46.50 | **85.87** |
| **AnyTouch** | TAG, VisGel, Cloth, TVL, SSVTP, YCB-Slide, OF Real , Octopi, TacQuad | **49.62** | 76.02 |

Table 4: Evaluation on the real-world pouring task using GelSight Mini.

| Method | Dynamic Perception | Mean Error (g) ↓ | |
|---|---|---|---|
| | | Fine-tune | Freeze |
| CLIP | ✗ | 5.22 | 49.1 |
| T3 | ✗ | 2.33 | 9.74 |
| **AnyTouch** | ✗ | 2.45 | 9.60 |
| **AnyTouch** | ✔ | **1.56** | **8.22** |

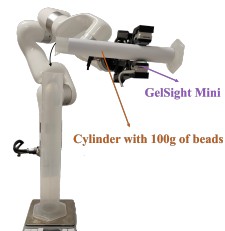

Figure 5: Setup of real-world pouring task.

are much simpler, and the tactile text descriptions in other datasets also include these two binary attributes, which have less impact on the data distribution.

### 5.5 STATIC PERCEPTION ON UNSEEN SENSORS (**Q3**)

To verify the generalization of our method on unseen sensors, we compare it with the multi-sensor models UniTouch and UniTouch† on two datasets from unseen sensors, OF 1.0 and OF 2.0. As shown in Table 3, the AnyTouch trained on the same data as Unitouch outperforms it on both datasets, demonstrating the static perception capability of our method across different sensors. Both UniTouch and AnyTouch, perform better on the unseen OF 1.0 dataset, confirming that integrating multi-sensor data aids generalization to unseen sensors. In addition, the AnyTouch trained on the full dataset achieves the highest performance on the unseen OF 1.0, demonstrating that learning sensor-agnostic semantic-level tactile features and constructing unified multi-sensor representation space is an effective approach for cross-sensor transfer.

### 5.6 REAL WORLD DYNAMIC PERCEPTION (**Q3**)

To test the dynamic perception capability of our method in real-world object manipulation tasks, we conduct experiments on a real-world task: fine-grained pouring, as shown in Figure 5. The detailed task setup is located at A.8. We compare AnyTouch with a recent multi-sensor model T3 and use a static-only AnyTouch model trained solely on tactile images as a baseline. Neither model includes modules specifically designed for dynamic perception. We conduct 10 real-world test runs and record the error between the poured mass and the target mass for each test. We then average the error across the test runs to get the "mean error" and use it as the metric, as shown in Table 4. When the tactile encoder is frozen and only the policy network is fine-tuned, CLIP, which has not seen tactile images, struggles with the task, highlighting the challenges of fine-grained dynamic perception. In contrast, the static-only AnyTouch performed comparably to T3 which was trained on more data. After integrating dynamic perception capabilities, AnyTouch achieved the best performance. This demonstrates the importance of learning unified multi-sensor representations from both static and dynamic perspectives for completing various tasks including real-world tasks.

### 6 CONCLUSION

In this paper, we collect TacQuad, an aligned multi-modal multi-sensor tactile dataset that enables the explicit integration of various sensors. Based on this, we present AnyTouch, a unified multi-sensor tactile representation learning framework from the perspectives of both static and dynamic perception. We also explore the multi-sensor representation space and sensor transferability.

ACKNOWLEDGMENTS

This project is sponsored by CCF-Zhipu.AI Large Model Innovation Fund and also supported by the Zhipu Company. We want to thank Xinjie Tang for providing design and drafting support.

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

Table 5: Training dataset statistics. The text modality in Touch and Go and ObjectFolder Real is generated by GPT-4o. *Note that Cloth and YCB-Slide contain vision modality, but we intentionally do not use it to demonstrate our method's compatibility with modality absence. We only count the number of contact frames used for training in each dataset.

| Dataset | Vision | Text | Video | Sensor | Size |
|---|---|---|---|---|---|
| Touch and Go (Yang et al., 2022) | ✔ | ✔ | ✔ | GelSight | 250k |
| VisGel (Li et al., 2019) | ✔ | ✘ | ✔ | GelSight | 587k |
| Cloth (Yuan et al., 2018) | ✘* | ✘ | ✔ | GelSight | 587k |
| TVL (Fu et al., 2024) | ✔ | ✔ | ✔ | DIGIT | 39k |
| SSVTP (Kerr et al., 2022) | ✔ | ✔ | ✘ | DIGIT | 4.5k |
| YCB-Slide (Suresh et al., 2023) | ✘* | ✘ | ✔ | DIGIT | 183k |
| ObjectFolder Real (Gao et al., 2023) | ✔ | ✔ | ✔ | GelSlim | 1165k |
| Octopi (Yu et al., 2024) | ✘ | ✔ | ✔ | GelSight Mini | 39k |
| TacQuad | ✔ | ✔ | ✔ | GelSight, DIGIT, DuraGel GelSight Mini | 55k |

# A APPENDIX

## A.1 TRAINING DATASET STATISTICS

In this section, we provide a detailed presentation of the sensor type, modality pairing and data scale of the datasets used during the training phase. We use a total of 9 datasets from 5 different sensors for training, including: Touch and Go (TAG) (Yang et al., 2022), VisGel (Li et al., 2019) and Cloth (Yuan et al., 2018) from GelSight (Yuan et al., 2017); ObjectFolder Real (OF Real) (Gao et al., 2023) from GelSlim (Donlon et al., 2018); TVL (Fu et al., 2024), YCB-Slide (Suresh et al., 2023) and SSVTP (Kerr et al., 2022) from DIGIT (Lambeta et al., 2020); Octopi (Yu et al., 2024) from GelSight Mini (Inc.); and the coarse-grained subset of our TacQuad from DIGIT, GelSight Mini and DuraGel (Zhang et al., 2024). We filter the contact frames with tactile deformations by calculating the difference between each tactile image and the corresponding background frame in these datasets. Eventually, we extract a total of 2,481,703 tactile contact frames from these datasets for model training. We also leverage the continual frames available in these datasets to train the model's dynamic perception capabilities. The detailed training dataset statistics are shown in Table 5. We generate text descriptions for Touch and Go and ObjectFolder Real using GPT-4o. We also expand the text descriptions in TVL, SSVTP and Octopi. *We remove the text modality of the training samples included in the test set of downstream tasks for fairness.* It is worth saying that Cloth and YCB-Slide contain vision modality originally, but we intentionally do not use it to demonstrate our method's compatibility with missing modalities.

## A.2 MULTI-MODAL ALIGNING LOSS

When training AnyTouch, we maximize the use of paired data by selecting the largest subset for each modality combination within the batch for multi-modal aligning. Considering a pair of uni-modal representations $(x_T, x_V, x_L)$ derived from uni-modal encoders, where $x_T \in \mathbb{R}^d$ is the touch representation, $x_V \in \mathbb{R}^d \cup \varnothing$ is the vision representation and $x_L \in \mathbb{R}^d \cup \varnothing$ is the text representation. We then perform multi-modal alignment (Radford et al., 2021) within the batch as:

$$
\begin{aligned}
\mathcal{L}_{T \to V} &= -\frac{1}{|\Omega_V|} \sum_{i \in \Omega_V} \log \frac{\exp(x_{T,i}^\top \cdot x_{V,i}/\tau)}{\sum_{j \in \Omega_V} \exp(x_{T,i}^\top \cdot x_{V,j}/\tau)}, \\
\mathcal{L}_{T \to L} &= -\frac{1}{|\Omega_L|} \sum_{i \in \Omega_L} \log \frac{\exp(x_{T,i}^\top \cdot x_{L,i}/\tau)}{\sum_{j \in \Omega_L} \exp(x_{T,i}^\top \cdot x_{L,j}/\tau)}, \\
\mathcal{L}_{V \to L} &= -\frac{1}{|\Omega_V \cap \Omega_L|} \sum_{i \in \Omega_V \cap \Omega_L} \log \frac{\exp(x_{V,i}^\top \cdot x_{L,i}/\tau)}{\sum_{j \in \Omega_v \cap \Omega_L} \exp(x_{V,i}^\top \cdot x_{L,j}/\tau)},
\end{aligned}
\tag{10}
$$

where $B$ is the batchsize, $\Omega_V, \Omega_L$ are sets of indices for the samples containing vision and text, and $\tau$ is the scalar temperature.

## A.3 DOWNSTREAM DATASETS

In this section, we provide a more detailed introduction to the downstream datasets. Specifically, we compare the static perception capabilities of AnyTouch and the baselines on four downstream datasets: TAG, Feel (Calandra et al., 2017), ObjectFolder 1.0 (OF 1.0) (Gao et al., 2022a) and OjectFolder 2.0 (OF 2.0) (Gao et al., 2022b). TAG includes three tactile properties understanding tasks: material, hardness, and roughness classification. Feel is a robotic dataset from GelSight containing a grasp success prediction task. We follow the data split in (Yang et al., 2024; Cheng et al., 2024) for Feel. ObjectFolder 1.0 and OjectFolder 2.0 are two simulated object datasets using TACTO (Wang et al., 2022a) and Taxim (Si & Yuan, 2022). We use them as unseen datasets from unseen sensors, and follow the data split in Yang et al. (2024).

## A.4 BASELINES

In the static perception task, we compared AnyTouch with several recent single-sensor and multi-sensor baselines. VIT-LENS-2 (Lei et al., 2024), TLV-Link (Cheng et al., 2024), and Omnibind (Lyu et al., 2024) are three single-sensor models included, all of which conduct multi-modal alignment using data from GelSight. As for multi-sensor models, we compare our method with UniTouch (Yang et al., 2024), which currently demonstrates the SOTA cross-sensor performance. We also intend to train UniTouch and TLV-Link using all available multi-sensor data for comparison. However, since UniTouch requires touch-vision paired data and TLV-Link can only be trained on three-modal paired data, we extract the largest subset of data that meets the requirements to train them, remarked as UniTouch† and TLV-Link†.

In the real-world dynamic perception task, we compare with a recent multi-sensor model T3 (Zhao et al., 2024). This model has been validated to have strong capabilities in completing manipulation tasks. It is important to note that the amount of training data used by this model is approximately 3M, which is more than our AnyTouch. Additionally, since this model utilizes labels from various downstream tasks during pre-training, comparing its static perception capabilities with other models would be unfair. Therefore, we only use it in the dynamic perception task.

## A.5 DETAILS FOR FINE-GRAINED DATA COLLECTION

In this section, we provide a more detailed introduction of the data collection process for the fine-grained spatio-temporal aligned data. The calibration platform we built consists of three main parts: a platform, a movable end effector, and a 3D-printed container that holds the sensor. The four sensors are fixed side by side in the container. The movable end on our calibration platform can be programmed to move at a specified speed to a designated position within the coordinate system defined by the base. Therefore, as long as we pre-measure the relative positions of the centers of the four sensor surfaces within the container and compensate for the relative positions during each set of data collection, we can ensure that all four sensors make contact with the object from the same initial position and at the same speed, thereby achieving both temporal and spatial alignment.

## A.6 IMPLEMENTATION DETAILS

We base our encoders on OpenCLIP-Large (Cherti et al., 2023). For the tactile decoder, we use a Vision Transformer (ViT) (Dosovitskiy et al., 2020) with 8 layers and a dimension of 512. We use the AdamW (Loshchilov, 2017) optimizer with a learning rate of 2e-4. After a warm-up period of 1 epoch, we implement linear learning rate decay. For each tactile video clip, we use $T = 3$ frames. We train the first stage for 20 epochs and the second stage for 12 epochs on 4 NVIDIA A800 GPUs. We alternate between training with tactile images and video clips throughout the entire training process. We use a mask ratio $\rho = 0.75$. During the alignment, we use the text modality as the anchor, freezing the text encoder while performing LoRA fine-tuning on the vision encoder. We set the alignment strength $\alpha_{TV} = \alpha_{TL} = 1.0$ and $\alpha_{VL} = 0.2$, and set the weight of cross-sensor matching $\lambda = 0.1$. Following (Yang et al., 2024), we use $L = 5$ sensor tokens for each type of sensor. In both stages, we set the probability of using universal sensor tokens $p_u$ to increase linearly from 0 to 0.75.

Focus on the object being touched by the handheld tactile sensor. Please provide a description of the tactile properties (exact location of the touch, materials at the contact point, texture, roughness level, hardness level, and more than these properties), using a series of adjectives. Please do not use ambiguous adjectives like 'moderate' or 'medium'. Ensuring your response does not exceed 60 words, counting punctuation as separate words.

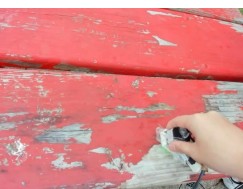

**Output**: The sensor touches a rough, splintered wooden bench with peeling paint. Textured surface, visibly cracked, and fragmented. Hardness level high, unevenness significant, and the material at the contact point is weathered, aged wood coated in deteriorating paint chips.

Figure 6: **Prompt and raw output for TAG and TacQuad.**

### A.7 GPT-4O ANNOTATION

In this work, we generate paired text descriptions of tactile properties for Touch and Go, Object-Folder Real and our TacQuad. We input paired visual images and predefined text prompts into GPT-4o to obtain text descriptions. We borrow the prompt from (Cheng et al., 2024) and make appropriate adjustments. The prompt for Touch and Go and TacQuad is shown in Figure 6.

Specifically, because the ObjectFolder Real dataset has two camera views and some touch locations or details may not be visible, we input two visual images and one tactile image simultaneously, with clear indications in the prompt, as shown in Figure 7.

Since the TVL and SSVTP datasets only contain simple phrase-level tactile descriptions, we also use GPT-4o to extend the text modality in both datasets. We input both visual images and the existing text descriptions simultaneously, as shown in Figure 8.

After generating the annotations using GPT-4o, we conduct a simple and rough sampling check to ensure the correctness. We find that as long as the model can correctly recognize the object, the knowledge of object-related physical properties stored in the large model can effectively provide accurate tactile annotations. Errors in the generated annotations may occur when an object's category is difficult to determine or when occlusions are present. However, such instances are relatively uncommon.

### A.8 REAL-WORLD POURING TASK

To test the dynamic perception capability of our method in real-world object manipulation tasks, we conduct experiments on a real-world task: fine-grained pouring, as shown in Figure 5. In the experiments, we use a 6-DoF UFACTORY xArm 6 robotic arm equipped with a Robotiq 2F-140 gripper. Cartesian space displacement commands are generated at a policy frequency of 5 Hz. The robot arm must rely entirely on tactile feedback to pour out 60g of small beads from a cylinder that initially contains 100g of beads. The robot arm can select one of the three actions to perform based on the real-time tactile feedback: pouring, waiting, or retracting. The action step size is $\delta\phi = 0.25°$. We train the model through imitation learning and collect the training data using a keyboard. As both rotating the cylinder and pouring out the small beads lead to continuous variations in pressure on the sensors, the model must analyze the fine-grained changes between tactile images to determine the appropriate pouring speed and the right moment to retract the cylinder. This task is typically performed using multi-modal data (Li et al., 2022), making it particularly challenging for models that rely solely on tactile perception.

Focus on the object being touched by the tactile sensor in the first two images. The two images are shot from two distinct viewpoints. The third image comes from the tactile sensor, reflecting the detailed deformations when touching the object. Please provide a description of the tactile properties (exact location of the touch, materials at the contact point, texture, roughness level, hardness level, and more than these properties), using a series of adjectives. Please do not use ambiguous adjectives like 'moderate' or 'medium'. Ensuring your response does not exceed 60 words, counting punctuation as separate words.

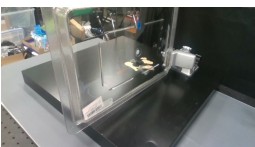 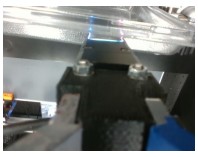 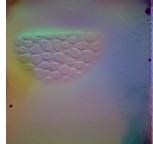

**Output without tactile image**: The tactile sensor contacts the upper center of a clear polycarbonate box lid. The material is smooth, glossy, and transparent, featuring a hard, durable surface. The texture is sleek, with no visible roughness or irregularities, providing a nearly frictionless feel The hardness level is high, indicative of robust plastic material.

**Output with tactile image**: The tactile properties indicate contact at the smooth, transparent polycarbonate lid, characterized by its high hardness and rigidity. **The surface is designed with a bubble-like texture that feels slightly raised and evenly distributed.** The texture appears consistent, lacking significant roughness, while retaining a firm yet slightly flexible quality at the touchpoint.

Figure 7: **Prompt and raw output for OF Real.** Given that OF Real includes two camera views and some touch locations or details may be obscured, we input two visual images along with one tactile image simultaneously. If the tactile image is not provided as input, there is a possibility of producing incorrect annotations (marked in red).

Focus on the object being touched by the tactile sensor. Please provide a description of the tactile properties (exact location of the touch, materials at the contact point, texture, roughness level, hardness level, and more than these properties), using a series of adjectives. It is already known that the object is {phrase-level descriptions}. Please do not use ambiguous adjectives like 'moderate' or 'medium'. Ensuring your response does not exceed 60 words, counting punctuation as separate words.

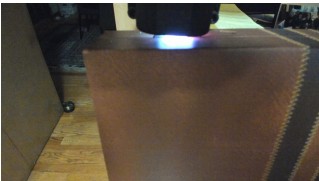

**Output**: The touch is at the upper edge, the tactile sensor contacts leather. The texture is smooth, with a slight grain. The roughness is low, and the hardness is low. The material feels soft, yielding minimally to pressure, with a consistent, uninterrupted surface.

Figure 8: **Prompt and raw output for TVL and SSVTP.** Given that these datasets only contain phrase-level tactile descriptions, we input the visual image and the phrase-level descriptions to generate more detailed tactile descriptions.

## A.9 ABLATION STUDY

To investigate the impact of each module in AnyTouch, as well as the individual contributions of the vision and text modalities in multi-modal alignment, we conduct ablation studies on the four downstream datasets. The experimental results are shown in Table 6. We observe performance decline when the paired vision and text modalities are excluded, highlighting the importance of aligning with these paired modalities to narrow the sensor gaps and achieve a comprehensive tactile perception capability. We also find that the performance decline caused by removing the visual modality is greater than removing text. However, this does not necessarily indicate that the visual modality is more important, as removing the visual modality results in a more significant reduction

Table 6: The impact of modalities and modules in AnyTouch on static perception capabilities.

| Model | TAG
Material | Feel
Grasp | OF 1.0
Material | OF 2.0
Material |
|---|---|---|---|---|
| **AnyTouch** | **80.82** | **80.53** | **49.62** | **76.02** |
| w/o Text Modality | 75.91(↓4.91) | 78.93(↓1.60) | 48.87(↓0.75) | 75.52(↓0.50) |
| w/o Vision Modality | 74.55(↓6.27) | 77.30(↓3.23) | 48.12(↓1.50) | 75.22(↓0.80) |
| w/o Text in TacQuad | 80.70(↓0.12) | 80.19(↓0.34) | 49.21(↓0.41) | 75.91(↓0.11) |
| w/o Stage 1 | 78.34(↓2.48) | 78.62(↓1.91) | 48.75(↓0.87) | 76.08(↑0.06) |
| w/o Stage 2 | 68.64(↓12.18) | 72.39(↓8.14) | 46.50(↓3.12) | 73.09(↓2.93) |
| w/o Cross-Sensor Matching | 80.54(↓0.28) | 79.43(↓1.10) | 49.25(↓0.37) | 75.80(↓0.22) |
| w/o Dynamic Perception | 77.93(↓2.89) | 79.28(↓1.25) | 48.62(↓1.00) | 75.70(↓0.32) |
| w/o Universal Sensor Tokens | 80.79(↓0.03) | 79.03(↓1.53) | 48.40(↓1.22) | 75.40(↓0.62) |

in data during the aligning. We also remove the text from the TacQuad dataset we proposed to validate the effectiveness of the text in the dataset. Although the TacQuad data is relatively small compared to the total dataset, making it unlikely to significantly impact performance when modified, we observe a consistent decline in the model's performance on downstream tasks after removing the text modality. This demonstrates the important role of the text modality in our dataset as a bridge that helps reduce the gap between sensors.

When we remove cross-sensor matching and universal sensor tokens from AnyTouch, we observe a performance decline primarily on the datasets from unseen sensors. It is important to note that the sensors in these datasets are not included in the positive sample pairs for cross-sensor matching, indicating that this task has even greater potential. This demonstrates that both strategies can enhance the model's generalization to unseen sensors. After removing the entire stage 2, we observe a significant performance decline. On OF 2.0, it performs even worse than CLIP, which has never encountered tactile data. This result is consistent with the improvement on OF 2.0 when removing stage 1 and the analysis in Section 5.3, indicating that learning semantic-level features is crucial for achieving comprehensive tactile perception and cross-sensor generalization. Nevertheless, learning pixel-level features in stage 1 is still meaningful for the seen sensors. In addition, we also observe a consistent decline in performance after removing the joint training for dynamic perception, indicating that integrating dynamic perception can indeed enhance static perception capabilities.

### A.10 CROSS-SENSOR GENERATION

To more comprehensively demonstrate the value and impact of the dataset we proposed, we conduct cross-sensor generation experiments on the fine-grained spatio-temporal aligned data. Specifically, we trained models to generate aligned DuraGel images from GelSight Mini images, and to reconstruct the 20x20 force fields captured by the Tac3D sensor from DIGIT and GelSight Mini data. We compared the performance of our model with the T3 model, which used more training data than ours (3.08M compared to our 2.48M) for pretraining. Specifically, for generating Duragel images, we constructed a GAN network based on ViT, using T3 or AnyTouch as the encoders for the discriminator and generator, similar to ViTGAN (Lee et al., 2022). A ViT-based decoder is then used to generate images across sensors. For the force field generation of Tac3D, due to its low resolution, we treat it as a regression task and use an MLP to reconstruct the force field based on the features extracted by the encoder. Both networks can effectively evaluate the quality of the encoder's tactile representations. To further ensure fairness, we also removed the overlapping portions of the coarse-grained aligned data from the training data that overlapped with this dataset. Note that Tac3D is an unseen sensor for both of the models. We use mean square error (MSE) (↓) between the generated data and the ground truth as the metric. The results shown in Table 7 indicate that our method outperforms T3 in terms of generation quality, both for cross-sensor generation of vision-tactile images and for force fields captured by the unseen Tac3D. This demonstrates the effectiveness of our method and the value of the dataset, and supports our motivation to obtain a unified tactile multi-sensor representation that is applicable to a variety of tasks and sensors.

Table 7: Performance comparison with T3 on the cross-sensor generation task using the fine-grained spatio-temporal aligned data.

| Model | Training Data | Mean Square Error ($\downarrow$) | | |
|---|---|---|---|---|
| | | GelSight Mini $\rightarrow$ DuraGel | GelSight Mini $\rightarrow$ Tac3D | DIGIT $\rightarrow$ Tac3D |
| T3 | 3.08M | 0.2261 | 0.0167 | 0.0155 |
| **AnyTouch** | 2.48M | **0.2159** | **0.0151** | **0.0144** |

## A.11 DISCUSSION ON FRAME NUMBER

In the real world, the complete process of touching an object can take several seconds or even tens of seconds. Ensuring the model can comprehend an entire tactile video presents a significant challenge. Current large-scale video understanding models, such as Video-LLaMA (Zhang et al., 2023), often process tens or even hundreds of frames as input, encoding them into tokens. However, this comes at the cost of generating very long token sequences, which significantly increase computational overhead and inference time. The tactile modality is frequently used in fine-grained manipulation tasks that demand high real-time performance, which imposes strict requirements on the model's inference speed. As a result, models that rely on long frame sequences are challenging to apply in real-time dynamic perception tasks. Moreover, since touch actions are typically performed at high speeds, even a sequence of three continual frames (equivalent to 0.1 seconds for a DIGIT sensor with a frequency of approximately 30Hz) can exhibit noticeable changes. We anticipated these challenges and, as a result, chose to use a sequence of three continual frames as the input format for tactile videos. This approach also enables the understanding of longer videos by selecting multiple 3-frame segments and either concatenating or summing their features, similar to ImageBind (Girdhar et al., 2023). Using more frames may lead to better perception performance, but this is essentially a trade-off between performance and both computational cost and reference speed.

## A.12 DISCUSSION ON OTHER TACTILE SENSORS

Tactile perception is not limited to images. Some tactile properties, such as temperature and torque, are difficult to obtain from tactile images alone, requiring the use of other types of tactile sensors. This issue presents challenges from both hardware and algorithmic perspectives.

From a hardware perspective, an ideal tactile sensor should be capable of gathering various types of tactile information, effectively integrating multiple existing tactile sensors into a single unit. This may be very challenging, and a more practical solution might involve equipping different fingers of a robotic hand with different types of sensors. This would allow for the simultaneous collection of diverse tactile data, maximizing the range of information captured.

From an algorithmic perspective, when vision-based tactile sensors are replaced with other types of tactile sensors (e.g., tactile sensor arrays), the multi-sensor data alignment method proposed in this paper can still be applied. Aligned data can then be used to perform alignment or to distill knowledge from the visuo-tactile model to models for other types of tactile sensors. For lower-resolution tactile sensors, the aligned data can facilitate tactile super-resolution learning, enabling knowledge transfer from vision-based tactile sensor models to enhance their performance.

If both vision-based tactile sensors and other tactile sensors (*e.g.*, those capturing temperature or other non-visual properties) are used simultaneously, a possible approach is to fuse their outputs into a unified, comprehensive tactile feature. This enriched representation can then be aligned with other modalities in a unified manner.

## A.13 LIMITATIONS AND FUTURE WORK

In this section, we discuss some potential limitations of our work and propose corresponding solutions for future work:

- **Compared to all the training data, the scale of the TacQuad dataset we have currently collected is still somewhat limited.** Capturing the immense variety of object types within

a single dataset is challenging in a limited amount of time. Fortunately, the coarse-grained spatial alignment data collection method we propose has the potential to scale up, as data collection can be performed manually without the need for precise alignment. Fine-grained data collection can also be expanded by replicating the calibration platform and increasing manpower. We plan to grow our team to scale up the dataset and enhance object diversity in future work.

- **The types of sensors considered are relatively limited.** We have made every effort to collect all available vision-based tactile sensors around us, yet we were only able to include four different types. Moreover, we did not explore the differences between individual sensors of the same type or address issues such as gel damage. Moving forward, we aim to expand our dataset and increase the diversity of sensors through collaborative data collection across multiple laboratories.

- **The scope of tasks for dynamic tactile perception is currently limited.** In this work, we validated the dynamic perception capabilities of our model on a single real-world manipulation task: pouring. We hope to explore more challenging and interesting dynamic perception tasks in future work. Additionally, beyond real-world manipulation tasks, studying tactile video understanding—particularly fine-grained dynamic tactile understanding that includes direction and action descriptions—is also an interesting direction to explore.

