# OpenReview forum: "AnyTouch: Learning Unified Static-Dynamic Representation across Multiple Visuo-tactile Sensors"
_ICLR.cc/2025/Conference — ICLR 2025 Poster_

### Official Review · Reviewer_oW9a · 2024-10-31

**Soundness:** 4
**Presentation:** 4
**Contribution:** 4
**Rating:** 8
**Confidence:** 4

**Summary:**

This paper present a novel framework called UltraTouch and a large dataset TacQuad collected using 4 different tactile sensors. The UltraTouch framework has a multi-stage training in pixel level and semantic level. In stage 1, the authors use MAE to train on static and dynamic tactile images/videos. In stage 2, the authors do contrastive learning on multi-modal aligning and cross-sensor matching to understand semantic-level sensor-agnostic tactile properties. The authors also answer research questions by testing on downstream tasks like material classification, grasp prediction, and pouring.

**Strengths:**

Strengths of this paper includes:

- They collect and open-source a large tactile dataset called TacQuad, which makes good contribution to the multi modal robot learning community since the tactile data are far from enough compared with vision data.
\
\
- The framework is novel. The idea of learning a unified tactile representation is very interesting and the authors study how tactile aligns with other modalities like vision and text, and how different sensors match with each other.
\
\
- The experiments are comprehensive. The authors conduct extensive experiments on multiple downstream tasks like material classification, grasp prediction, and pouring, which demonstrate Ultratouch, learning unified representations, has advantages over other models.

**Weaknesses:**

Weaknesses of this paper include:

- Though vision based tactile sensors are very popular in robot learning community, it might not be the ultimate solution that people agree on. Tactile perception are more than images. Humans feel temperature, texture, shape, vibrations, etc. when interacting with objects. This is still an open question and is out of scope of this paper. But it will be interesting to see if how other tactile sensors align or match with the vision-based tactile sensors.
\
\
- I think there can be some study on the effect from the stage 1, the pixel level learning, on the downstream tasks. How does learning the pixel-level details help the downstream tasks? I think material classification and grasp prediction might not need that level of detail to achieve good performance.

**Questions:**

- In data collection process, how do you localize the contact position in world coordinate? How to ensure the contact position is the same for different sensors?

- How many corrections do you have to make when use GPT to generate descriptions? How well does it understand the data?

- Are you using the same sensor throughout the entire data collection process? I feel like most tactile sensors now are not very robust, especially the gel pad, which will degrade over time when it touches objects for too many times. Have you replaced gels during the collection process? Will the data be different after replacement?

- Even the tactile sensors are of the same type. The difference between each individual sensor is huge. See the pixel-wise difference of DIGIT in Fig 7. in this paper [1]. Combined with my previous question, if you have use multiple sensors of the same type during collection and training, I'm curious how this will affect the performance.

- Missing space in line 797 between TAG and includes

[1] Bhirangi, Raunaq, et al. "AnySkin: Plug-and-play Skin Sensing for Robotic Touch." arXiv preprint arXiv:2409.08276 (2024).

---

> ### Author Response · Authors · 2024-11-24
>
> We would like to express our sincere appreciation for your insightful comments! We are very excited to receive your encouraging recognition of our "good contribution to the community", "novel framework" and "comprehensive experiments".
>
>
>
> > Though vision based tactile sensors are very popular in robot learning community, it might not be the ultimate solution that people agree on. Tactile perception are more than images. Humans feel temperature, texture, shape, vibrations, etc. when interacting with objects. This is still an open question and is out of scope of this paper. But it will be interesting to see if how other tactile sensors align or match with the vision-based tactile sensors.
>
> Thank you for the insightful suggestion. We agree that tactile perception is not limited to images. Some tactile properties, such as temperature and torque, are difficult to obtain from tactile images alone, requiring the use of other types of tactile sensors. This issue presents challenges from both hardware and algorithmic perspectives.
>
> From a hardware perspective, an ideal tactile sensor should be capable of gathering various types of tactile information, effectively integrating multiple existing tactile sensors into a single unit. This may be very challenging, and a more practical solution might involve equipping different fingers of a robotic hand with different types of sensors. This would allow for the simultaneous collection of diverse tactile data, maximizing the range of information captured.
>
> From an algorithmic perspective, when vision-based tactile sensors are replaced with other types of tactile sensors (e.g., tactile sensor arrays), the multi-sensor data alignment method proposed in this paper can still be applied.  Aligned data can then be used to perform alignment or to distill knowledge from the visuo-tactile model to models for other types of tactile sensors. For lower-resolution tactile sensors, the aligned data can facilitate tactile super-resolution learning, enabling knowledge transfer from vision-based tactile sensor models to enhance their performance.
>
> If both vision-based tactile sensors and other tactile sensors (e.g., those capturing temperature or other non-visual properties) are used simultaneously, a possible approach is to fuse their outputs into a unified, comprehensive tactile feature. This enriched representation can then be aligned with other modalities in a unified manner.
>
> We have included the related discussion in the revised version. Thank you again for your insightful comments!
>
> > I think there can be some study on the effect from the stage 1, the pixel level learning, on the downstream tasks. How does learning the pixel-level details help the downstream tasks? I think material classification and grasp prediction might not need that level of detail to achieve good performance.
>
> Thank you for raising this point. **In fact, we have already provided a detailed analysis of how the modules and the two stages influence the performance of our method in Table 6 of the appendix.** By comparing the rows removing stage 1 and stage 2, we observe that learning fine-grained tactile details can indeed improve performance on the TAG (material) and Feel (grasp) datasets. However, this improvement is much smaller than the gain achieved by learning coarse-grained tactile features that are more closely related to the tasks. Thank you again for your comment.
>
> > In data collection process, how do you localize the contact position in world coordinate? How to ensure the contact position is the same for different sensors?
>
> Thank you for raising this important point. The movable end on our calibration platform can be programmed to move at a specified speed to a designated position within the coordinate system defined by the base. Therefore, as long as we pre-measure the relative positions of the centers of the four sensor surfaces within the container and compensate for the relative positions during each set of data collection, we can ensure that all four sensors make contact with the object from the same initial position and at the same speed, thereby achieving both temporal and spatial alignment. We have provided a more detailed explanation of the data collection in the revised version. Thank you again for your reminder!

---

> > ### Author Response · Authors · 2024-11-24
> >
> > > Are you using the same sensor throughout the entire data collection process? I feel like most tactile sensors now are not very robust, especially the gel pad, which will degrade over time when it touches objects for too many times. Have you replaced gels during the collection process? Will the data be different after replacement?
> >
> > Thank you for raising this point. We used the same sensor for each type throughout the entire data collection process. Since most of our data collection was done handheld, and we did not collect very sharp objects, we did not encounter the issue of gel damage. However, we also agree that the robustness of vision-based tactile sensors is an issue that needs to be addressed. Among all tactile sensors, the DIGIT sensor specifically addresses the robustness issue by improving the manufacturing process of the gel, making it harder. For other sensors that use softer gels, there is indeed a possibility of gel damage occurring during data collection. We think that gel damage could impact tactile perception performance, as it introduces noise that the model has not been exposed to before. Finding ways to address this issue from the perspective of data or model design could be an interesting direction for future work. However, since the gel of the same sensor typically does not show significant shape differences, we believe that the replacement will not change the data collected. Thank you again for your valuable questions!
> >
> >
> >
> > > Even the tactile sensors are of the same type. The difference between each individual sensor is huge. See the pixel-wise difference of DIGIT in Fig 7. in this paper [1]. Combined with my previous question, if you have use multiple sensors of the same type during collection and training, I'm curious how this will affect the performance.
> >
> > Thank you for your insightful comments. We agree that differences may also exist between individual sensors of the same type. For example, the DIGIT sensor, which uses three RGB LEDs for internal illumination, inherently allows adjustable lighting intensity [1], leading to variations between sensors. We also believe that using multiple sensors of the same type for data collection can enhance the diversity of the dataset and benefit model training, as it can be considered a form of data augmentation. However, this approach would be prohibitively expensive and time-consuming, requiring a larger number of sensors and more personnel for data collection. As our data collection method is highly scalable, enabling collaborative data collection across multiple researchers and laboratories, and we aim to further enhance the diversity and scale of the dataset in future work. Thank you again for your constructive suggestions!
> >
> >
> >
> > [1] Mike Lambeta, Po-Wei Chou, Stephen Tian, Brian Yang, Benjamin Maloon, Victoria Rose Most, Dave Stroud, Raymond Santos, Ahmad Byagowi, Gregg Kammerer, et al. Digit: A novel design for a low-cost compact high-resolution tactile sensor with application to in-hand manipulation. IEEE Robotics and Automation Letters, 5(3):3838–3845, 2020.
> >
> >
> >
> > > Missing space in line 797 between TAG and includes
> >
> > Thank you very much for your careful reading! We have corrected these errors in the revised version.

---

### Official Review · Reviewer_dEU8 · 2024-11-03

**Soundness:** 2
**Presentation:** 3
**Contribution:** 3
**Rating:** 6
**Confidence:** 4

**Summary:**

This paper presents a novel and comprehensive representation learning framework for tactile data that addresses (1) V-T-L modalities alignment, (2) capturing dynamic information in tactile modality, and (3) generalization across different sensors together. It constructs a multi-sensor dataset called TacQuad and a 2-stage multi-task representation learning framework, equipped with pixel-level MAE, semantic-level multi-modal alignment, and cross-sensor matching techniques. With extensive experiments, the authors show this representation helps in some downstream tasks under various settings (seen and unseen sensors, different training dataset combinations, and real robot experiments).

**Strengths:**

1. Well-motivated and well-written. Authors observed the difference between tactile sensing and vision, and creatively designed corresponding modules to enhance the representation.

2. Clear visualization. Fig 2 works as a nice abstract of the entire framework.

**Weaknesses:**

1. In Tables 2 and 3, the average performance of the TAG+Feel+YCB-slide+OF 2.0 model is better than the full model with the TacQuad dataset. This observation
  - Questions the necessity of your TacQuad dataset, and
  - Poses a new question: how do you choose the size & scope of the dataset? Contrastive learning has requirements on the dataset size, while too much data will, in turn, make it "lose focus".

Please add more discussion and experiments on this issue, and figure out either the detailed reason (i.e. which dataset is bad for some reason) or how to find a good combination of datasets.

2. Given you claimed the model has the ability to extract sensor-agnostic features from unseen sensors, it would be more convincing to show some downstream task performance on the new sensor, aside from qualitatively showing t-SNE visualization.

3. Some other results are not convincing enough as well. In Fig 4 / Sec 5.3, DuraGel samples are better clustered than baselines, but some samples from GelSight and DIGIT, which are previously seen, are far from each other and mixed with other samples. Is it due to some difficulties in tuning the corresponding weight for loss?

**Questions:**

1. Regarding Fig 7 and 8, is there a significant gap between GPT-4o outputs for OF-Real and TVL/SSVTP, given you didn't use tactile data for the latter? If so, will this affect your training? If not, do you think GPT-4o is making use of tactile input, given it may not be capable of analyzing images from tactile sensors? Can you show some raw output examples?

2. I noticed your "full model" removed Feel and OF 2.0 from the dataset. Is it on purpose? If the full model sees the Feel dataset, will its performance in Table 2 be lower than 80.53, between 80.53 and 82.3, or higher than 82.3? Similarly for OF 2.0. The claim that more is not always better itself is interesting, but I just wonder can it be simply due to the lack of some specific dataset? Your current experiment setting cannot rule out this possibility.

3. Sec 5.4 explained the performance drop on material classification, but what about grasp?

4. The tables are a bit hard to read. Is the first row in Table 1 from CLIP as explicitly said in Tables 2 and 3? In Table 2, the colorful detailed dataset configuration makes it hard to find what to compare before carefully reading Sec 5.4.

Minor:
1. Citation format. Parenthesis missing in lots of scenarios.
2. Line 797 missing space.

---

> ### Author Response · Authors · 2024-11-24
>
> We want to thank you for the helpful comments and valuable feedback! We appreciate the comments that the paper is well-motivated and well-written, and the proposed method is creative.
>
> > Given you claimed the model has the ability to extract sensor-agnostic features from unseen sensors, it would be more convincing to show some downstream task performance on the new sensor, aside from qualitatively showing t-SNE visualization.
>
> Thank you for your valuable comments! **In fact, we have already compared and discussed the performance of our model on unseen sensor datasets in Section 5.5**. We conducted comparisons with the previous SOTA multi-sensor model, UniTouch, on two unseen sensor datasets: ObjectFolder 1.0 and ObjectFolder 2.0. As shown in Table 3, the UltraTouch trained on the same data as Unitouch outperforms it on both datasets, demonstrating the static perception capability of our method on unseen sensors.
>
> To further validate our model's ability to extract sensor-agnostic features and demonstrate the value of our proposed dataset, we trained the model on our collected fine-grained spatio-temporal aligned data for cross-sensor generation. Specifically, we trained models to generate the 20x20 force fields captured by the Tac3D sensor from DIGIT and GelSight Mini data. We compared the performance of our model with the T3 model, which used more training data than ours (3.08M compared to our 2.48M) for pretraining. We treat this task as a regression task and use an MLP to reconstruct the force field based on the features extracted by the encoder. To further ensure fairness, we also removed the overlapping portions of the coarse-grained aligned data from the training data that overlapped with this dataset. **Note that Tac3D is an unseen sensor for both of the models**. We use mean-square error (MSE) (↓) between the generated data and the ground truth as the metric. The results are as follows:
>
> Table II: Performance comparison with T3 on the cross-sensor generation task using the fine-grained spatio-temporal aligned data (MSE as metric).
>
> | Method     | Training Data | GelSight Mini -> Tac3D | DIGIT -> Tac3D |
> | ---------- | ------------- | ---------------------- | -------------- |
> | T3         | 3.08M         | 0.0167                 | 0.0155         |
> | UltraTouch | 2.48M         | **0.0151**             | **0.0144**     |
>
> The results show that our method outperforms T3 in terms of generation quality. This demonstrates the superior ability of our model to extract sensor-agnostic features. This supports our motivation to obtain a unified tactile multi-sensor representation that is applicable to a variety of tasks and sensors. We also look forward to further exploring a wider range of task types in future work. In particular, using unseen sensors for dynamic perception tasks could be an interesting challenge. We have added the experimental results and related discussions to the revised version. Thank you again for your constructive suggestions!
>
> > Regarding Fig 7 and 8, is there a significant gap between GPT-4o outputs for OF-Real and TVL/SSVTP, given you didn't use tactile data for the latter? If so, will this affect your training? If not, do you think GPT-4o is making use of tactile input, given it may not be capable of analyzing images from tactile sensors? Can you show some raw output examples?
>
> Thank you for your valuable questions. We additionally input tactile images when using GPT-4o to annotate the OF Real dataset because the visual images in the OF Real dataset were captured by two cameras positioned relatively far from the objects. As a result, the camera viewpoints are sometimes obstructed or fail to capture the tactile details at the contact points effectively. As illustrated in Figure 7 of the appendix, the images captured by both cameras fail to reveal the protrusions on the lid's surface. Without the guidance of tactile images, GPT-4o might incorrectly assume that the contact point on the lid’s surface is smooth, resulting in inaccurate annotations. This issue is not present in other datasets, allowing high-quality annotations to be achieved using only visual images. **Therefore, there is no significant difference between the annotations of the OF Real dataset and those of other datasets**. During the annotation process of the OF Real dataset, we found that if the collecting method and principles behind the tactile images are clearly outlined in the prompt, GPT-4o can grasp the concept of tactile images to some extent, even if it has never encountered them before. This is because GPT-4o can interpret images, and data from vision-based tactile sensors are essentially high-quality images. We have added the corresponding raw outputs from GPT-4o for all the prompt examples in the appendix in the revised version. Thank you again for your valuable questions!

---

> > ### Author Response · Authors · 2024-11-24
> >
> > > In Tables 2 and 3, the average performance of the TAG+Feel+YCB-slide+OF 2.0 model is better than the full model with the TacQuad dataset. This observation
> > >
> > > - Questions the necessity of your TacQuad dataset, and
> > > - Poses a new question: how do you choose the size & scope of the dataset? Contrastive learning has requirements on the dataset size, while too much data will, in turn, make it "lose focus".
> >
> > Thank you for your valuable comments. In Table 2 and 3, the performance of the TAG+Feel+YCB-slide+OF 2.0 model is better than the full model with the TacQuad dataset on TAG (material classification), Feel and OF 2.0. The phenomenon on Feel and OF 2.0 is easy to explain: **Feel and OF 2.0 dataset is included in the training data of the TAG+Feel+YCB-slide+OF 2.0 model, but not in the training data of the full model**. Therefore, it is not surprising that the performance of the TAG+Feel+YCB-slide+OF 2.0 model is better than the full model with the TacQuad dataset (but without Feel and OF 2.0) on Feel and OF 2.0.
> >
> > The TAG (material classification) dataset shows this phenomenon because **it is included in the pre-training data and its proportion in the full dataset changes**. Due to the scarcity of tactile datasets, existing methods [1,2,3] sometimes validate downstream datasets already included in the pre-trained data. In this setting, **when the downstream task's dataset has a larger proportion in the pre-training data, it is naturally more likely to perform better.** This phenomenon, noted in the CLIP paper (see Figure 17) [4], suggests that when the overlap between the downstream task dataset and the pre-training dataset is higher, the model may achieve better performance on that dataset. Therefore, integrating more data reduces the proportion of TAG data in pre-training, leading to a performance decline in material classification for the seen TAG dataset. Notably, the performance on the roughness and hardness classification tasks on TAG does not decline. This is because these binary classification tasks are much simpler, and the tactile text descriptions we generated for all datasets also include these two binary attributes, which have less impact on the data distribution.
> >
> > **We also want to emphasize that when introducing more multi-sensor data, our method shows an overall performance improvement on unseen datasets**.  The results are shown as follows:
> >
> > Table I: Performance changes on the unseen datasets when introducing more multi-sensor data.
> >
> > | Method     | Tactile Training Data                                        | Feel (Grasp)     | OF 1.0 (Material) | OF 2.0 (Material) |
> > | ---------- | ------------------------------------------------------------ | ---------------- | ----------------- | ----------------- |
> > | CLIP       | /                                                            | 72.37            | 41.00             | 73.16             |
> > | UltraTouch | TAG, VisGel, Cloth                                           | 79.12 (↑6.75)    | 46.12 (↑5.12)     | 75.10 (↑1.94)     |
> > | UltraTouch | TAG, VisGel, Cloth, OF Real                                  | 79.28 (↑0.16)    | 47.55 (↑1.43)     | 75.53 (↑0.43)     |
> > | UltraTouch | TAG, VisGel, Cloth, OF Real, TVL, SSVTP, YCB-Slide           | 79.10 (↓0.18)    | 48.00 (↑0.45)     | 75.57 (↑0.04)     |
> > | UltraTouch | TAG, VisGel, Cloth, OF Real, TVL, SSVTP, YCB-Slide, Octopi   | 79.40 (↑0.30)    | 48.75 (↑0.75)     | 75.66 (↑0.09)     |
> > | UltraTouch | TAG, VisGel, Cloth, OF Real, TVL, SSVTP, YCB-Slide, Octopi, **TacQuad** | **80.53**(↑1.13) | **49.62** (↑0.87) | **76.02**(↑0.36)  |
> >
> > Such datasets or downstream applications are actually more common and closer to real-world scenarios, as it is unlikely that we can fully encompass real-world data in the pre-training data. **This aligns with the core goal of our dataset and our method: to obtain a unified tactile multi-sensor representation that is applicable to a variety of tasks and sensors**.
> >
> > **To benefit from data in other datasets, a necessary condition is that there must be transferable overlap between these datasets**. For example, samples of the "wood" material from other datasets can assist in classifying "wood" in the downstream dataset. **However, a major issue with current tactile datasets is their limited object diversity**. For instance, the OF Real dataset, which contains up to 1,165k contact frames, only includes 100 objects from 7 different categories, and some of these categories are not even present in the TAG dataset (e.g., glass). Such datasets offer limited transferable knowledge for the seen material classification dataset. **This actually supports our goal of collecting data from a wider variety of everyday objects**. **Scaling up is not just about increasing data volume, but more importantly, improving object diversity**.
> >
> > We have expanded this discussion in the revised version. Thank you again for your valuable comments!

---

> > > ### Author Response · Authors · 2024-11-24
> > >
> > > [1] Fengyu Yang, Chao Feng, Ziyang Chen, Hyoungseob Park, Daniel Wang, Yiming Dou, Ziyao Zeng, Xien Chen, Rit Gangopadhyay, Andrew Owens, et al. Binding touch to everything: Learning unified multimodal tactile representations. In Proceedings of the IEEE/CVF Conference on Computer Vision and Pattern Recognition, pp. 26340–26353, 2024.
> > >
> > > [2] Weixian Lei, Yixiao Ge, Kun Yi, Jianfeng Zhang, Difei Gao, Dylan Sun, Yuying Ge, Ying Shan, and Mike Zheng Shou. Vit-lens: Towards omni-modal representations. In Proceedings of the IEEE/CVF Conference on Computer Vision and Pattern Recognition, pp. 26647–26657, 2024.
> > >
> > > [3] Yuanhuiyi Lyu, Xu Zheng, Dahun Kim, and Lin Wang. Omnibind: Teach to build unequal-scale modality interaction for omni-bind of all. arXiv preprint arXiv:2405.16108, 2024.
> > >
> > > [4] Alec Radford, Jong Wook Kim, Chris Hallacy, Aditya Ramesh, Gabriel Goh, Sandhini Agarwal, Girish Sastry, Amanda Askell, Pamela Mishkin, Jack Clark, et al. Learning transferable visual models from natural language supervision. In International conference on machine learning, pp. 8748–8763. PMLR, 2021.
> > >
> > > > Some other results are not convincing enough as well. In Fig 4 / Sec 5.3, DuraGel samples are better clustered than baselines, but some samples from GelSight and DIGIT, which are previously seen, are far from each other and mixed with other samples. Is it due to some difficulties in tuning the corresponding weight for loss?
> > >
> > > Thank you for asking this valuable question. In Figure 4, we illustrate the multi-sensor representation spaces of four different models (from left to right): (1) the CLIP model, which has not been exposed to any tactile data; (2) UltraTouch trained on a large amount of multi-sensor tactile data but only trained by the first stage (MAE); (3) UltraTouch trained with both the first stage and multi-modal alignment in the second stage (MAE + Align); and (4) the full UltraTouch model, which incorporates cross-sensor matching (MAE + Align + Match).
> > >
> > > In the CLIP model (on the far left), which has not been exposed to any tactile data, we observe that Duragel samples are more tightly clustered around the sensor compared to GelSight Mini and DIGIT samples. This is not an advantage, as these samples represent different tactile information and should, in fact, be more separated in the feature space. **In an ideal multi-sensor representation space, the representations of data from different tactile sensors should be clustered by the same object they represent, even though they come from different sensors, rather than having data from each sensor cluster around separate centers**. After multi-modal aligning and cross-sensor matching, the multi-sensor representation space of UltraTouch (on the far right) exhibits some of these characteristics. We can observe that many samples of the same color, including triangles (GelSight Mini samples), circles (DIGIT samples), and stars (DuraGel samples), cluster into small triangles connected by dashed lines, which is what we refer to as "clustering by objects". However, we also found that most of the same-colored circles (DIGIT samples) and triangles (GelSight Mini samples) are relatively close to each other, while the stars (DuraGel samples) are sometimes farther apart. This suggests that **it is more challenging to match Duragel images with GelSight Mini or DIGIT images that represent the same object**. We believe there are two main reasons for this: (1) The training data of the model includes DIGIT and GelSight Mini samples from other datasets, which results in a much larger amount of data from these sensors compared to the newly introduced DuraGel, leading to better representation quality. (2) The deformation features of the DuraGel sensor may not be as pronounced as those of other sensors, as GelSight Mini and DIGIT both show tri-color light changes at deformation points, while DuraGel only shows white light. Adjusting the loss weight when matching DuraGel samples with those from other sensors, or increasing the available data for DuraGel, are potential solutions. Thank you again for your valuable question!
> > >
> > > > The tables are a bit hard to read. Is the first row in Table 1 from CLIP as explicitly said in Tables 2 and 3? In Table 2, the colorful detailed dataset configuration makes it hard to find what to compare before carefully reading Sec 5.4.
> > >
> > > Thank you for your constructive suggestions. The first row in Table 1 is indeed from CLIP, as explicitly said in Tables 2 and 3. It represents the scenario where no tactile data is used for pre-training, and a linear probe is applied directly to the downstream task. We have refined the content and colors of the tables to enhance readability and help readers quickly grasp the information in the revised version. Thank you again for your suggestions!
> > >
> > >
> > >
> > > > 1. Citation format. Parenthesis missing in lots of scenarios.
> > > > 2. Line 797 missing space.
> > >
> > > Thank you for pointing these mistakes out! We have corrected them in the revised version.

---

> > > > ### Author Response · Authors · 2024-11-24
> > > >
> > > > > I noticed your "full model" removed Feel and OF 2.0 from the dataset. Is it on purpose? If the full model sees the Feel dataset, will its performance in Table 2 be lower than 80.53, between 80.53 and 82.3, or higher than 82.3? Similarly for OF 2.0. The claim that more is not always better itself is interesting, but I just wonder can it be simply due to the lack of some specific dataset? Your current experiment setting cannot rule out this possibility.
> > > >
> > > > Thank you for raising this important point. **We specifically removed Feel and OF 2.0 from the full dataset because we wanted to comprehensively evaluate and compare our model's performance on seen datasets from seen sensors, unseen datasets from seen sensors, and datasets from unseen sensors**. However, due to limitations in the baseline methods, they cannot train on the full dataset (UniTouch cannot use text, and TLV-Link requires all three modalities to be present). Additionally, the original paper for UniTouch used the unseen Feel and OF 2.0 datasets for our model. To ensure a fairer comparison, we use a smaller subset that includes Feel and OF 2.0 to train another UltraTouch model. As you suggested, we also trained an UltraTouch model using the full dataset that includes Feel and OF 2.0. The results are as follows:
> > > >
> > > > Table III: Performance comparison with models trained on different data.
> > > >
> > > > | Method     | Tactile Training Data                                        | Feel      | OF 2.0    |
> > > > | ---------- | ------------------------------------------------------------ | --------- | --------- |
> > > > | UniTouch   | TAG, Feel, YCB-Slide, OF2.0                                  | 82.3      | 85.4      |
> > > > | UltraTouch | TAG, Feel, YCB-Slide, OF2.0                                  | **87.17** | **85.87** |
> > > > | UltraTouch | TAG, VisGel, Cloth, TVL, SSVTP, YCB-Slide, OFReal, Octopi, TacQuad | 80.53     | 76.02     |
> > > > | UltraTouch | TAG, VisGel, Cloth, TVL, SSVTP, YCB-Slide, OFReal, Octopi, TacQuad, Feel, OF2.0 | 82.89     | 81.05     |
> > > >
> > > > We can observe that after introducing the Feel and OF 2.0 datasets, the model's performance on both datasets outperforms the UltraTouch model which has not been exposed to these two datasets. However, the performance of this model still does not outperform the UltraTouch model trained on the smaller subset. This is because these two datasets become seen datasets, and we have already analyzed the reason for the performance decline on the seen dataset when more data is introduced in the previous discussion. The main reason for this is that introducing more multi-sensor data lowers the proportion of the Feel and OF 2.0 data in the full dataset. We have added the experimental results and related discussions to the revised version. Thank you again for your insightful comments!
> > > >
> > > >
> > > >
> > > > > Sec 5.4 explained the performance drop on material classification, but what about grasp?
> > > >
> > > > Thank you for raising this important point. In Table 2, the Ultratouch model trained with a larger-scale multi-sensor dataset shows a performance drop on the grasp success prediction task of the Feel dataset compared to models trained with less data. This is because **the model trained with more data has not been exposed to the Feel dataset, while the latter has**, as indicated in the "Tactile Training Data" column of Table 2.
> > > >
> > > > When selecting downstream datasets, we aim to ensure task diversity and thoroughly evaluate our model's performance across various scenarios: seen and unseen datasets of known sensors, as well as datasets of entirely unseen sensors. Therefore, we selected the Feel dataset as an unseen dataset of seen sensors and intentionally excluded it from our model's training phase. However, since the existing UniTouch model incorporates the Feel dataset during training, we trained another UltraTouch model using the same data to ensure a fair comparison. Consequently, this version of UltraTouch achieved higher performance due to its exposure to the Feel dataset during training.
> > > >
> > > > We have clarified this point in the revised version. Thank you again for your comment!

---

### Official Review · Reviewer_TLvC · 2024-11-04

**Soundness:** 3
**Presentation:** 3
**Contribution:** 4
**Rating:** 8
**Confidence:** 4

**Summary:**

This paper proposes UltraTouch, a model that learns unified static-dynamic representations for multiple tactile sensors with different architectures. To train UltraTouch, a dataset called TacQuad is introduced by collecting multimodal data from four different tactile sensors. UltraTouch captures both pixel-level and semantic-level features, which are shown to be transferable and effective through experiments on multiple downstream tasks.

**Strengths:**

1. The collected dataset TacQuad, especially the fine-grained spatio-temporal aligned data can be a significant contribution to the community of tactile sensing. None of the prior works has collected tactile videos with shared speed and depth using multiple sensors. Although not fully explored in this work, I think this dataset and its data collection program will potentially benefit future research if being available to public.

2. The proposed UltraTouch framework, to the best of my knowledge, is the first model that learns the dynamic tactile representations through pretraining. This is a considerable improvement upon the prior works that learn unified tactile representations, which are all limited to static images. Future works that leverage touch for fine-grained manipulation tasks can potentially benefit from this framework.

3. The performance of UltraTouch model on the existing material classification/ grasp stability prediction seems promising, which is outperforming the previous pretraining frameworks significantly in most experiments.

4. The proposed real-world pouring task demonstrates the effectiveness of learning dynamic representations (Table 4).

**Weaknesses:**

1. As is shown in A.4, only 3 frames are used for each tactile video clip. This seems to be a surprisingly small number of frames, and my concern is that this might not be sufficient for extracting some of the fine-grained tactile information (e.g., hardness). One possible ablation experiment would be training the model with different video length and evaluate the performance. See Questions section for more details.

2. The paper proposes a novel calibration platform for collecting fine-grained spatio-temporal aligned data, and they claim that this can be used for fine-grained tasks such as cross-sensor generation. However, it seems that they don't explore much on the potential of this data, i.e., most of the downstream tasks are performed on static perception, and the only pouring task is not closely related to this part of data. It would be interesting to see if any more impressive tasks can be performed using the proposed dataset.

**Questions:**

1. Referring to my first concern in Weaknesses section, I'm curious about the computational cost of training with tactile videos. It's common that the frame rate of the video being limited due to data loading/ computation limits, but only using 3 frames still seems too few to me. My guess is that using more frames may lead to better dynamic perception ability, and I'm wondering if an ablation experiment on this parameter can be possible?

2. Besides TacQuad, some of the other datasets also contain tactile videos, such as TAG and OF Real. I wonder if they are already treated as videos during the training of UltraTouch, if not, maybe it can further improve the model performance if they are used as videos.

3. I'm curious about the scalability of the calibration platform used for data collection. Is it easy for other researchers to build a similar platform to scale up the current dataset?

---

> ### Author Response · Authors · 2024-11-24
>
> We want to thank you for the helpful comments and valuable feedback! We appreciate your recognition of the value of our dataset, the novelty of our framework and the comprehensive experiments.
>
>
>
> > As is shown in A.4, only 3 frames are used for each tactile video clip. This seems to be a surprisingly small number of frames, and my concern is that this might not be sufficient for extracting some of the fine-grained tactile information (e.g., hardness). One possible ablation experiment would be training the model with different video length and evaluate the performance. See Questions section for more details.
>
> > Referring to my first concern in Weaknesses section, I'm curious about the computational cost of training with tactile videos. It's common that the frame rate of the video being limited due to data loading/ computation limits, but only using 3 frames still seems too few to me. My guess is that using more frames may lead to better dynamic perception ability, and I'm wondering if an ablation experiment on this parameter can be possible?
>
> Thank you for your insightful comments. This is a valuable question, as in the real world, the complete process of touching an object can take several seconds or even tens of seconds. Ensuring the model can comprehend an entire tactile video presents a significant challenge. Current large-scale video understanding models, such as Video-LLaMA [1], often process tens or even hundreds of frames as input, encoding them into tokens. However, this comes at the cost of generating very long token sequences, which significantly increase computational overhead and inference time. The tactile modality is frequently used in fine-grained manipulation tasks that demand high real-time performance, which imposes strict requirements on the model's inference speed. As a result, models that rely on long frame sequences are challenging to apply in real-time dynamic perception tasks. Moreover, since touch actions are typically performed at high speeds, even a sequence of three continual frames (equivalent to 0.1 seconds for a DIGIT sensor with a frequency of approximately 30Hz) can exhibit noticeable changes. **We anticipated these challenges and, as a result, chose to use a sequence of three continual frames as the input format for tactile videos**. This approach also enables the understanding of longer videos by selecting multiple 3-frame segments and either concatenating or summing their features, similar to ImageBind [2]. We also agree that using more frames may lead to better perception performance, but this is essentially **a trade-off between performance and both computational cost and reference speed**. Since the current model which uses three frames as input, requires two days to complete training, and using more frames would take even longer, we are unable to provide results for an ablation study on the number of frames within a short time We have expanded this discussion in the revised version and we look forward to explore the related experiments in future work. Thank you again for your insightful comments!
>
> [1] Hang Zhang, Xin Li, and Lidong Bing. Video-llama: An instruction-tuned audio-visual language model for video understanding. In Proceedings of the 2023 Conference on Empirical Methods in Natural Language Processing: System Demonstrations, pp. 543–553, 2023.
>
> [2] Rohit Girdhar, Alaaeldin El-Nouby, Zhuang Liu, Mannat Singh, Kalyan Vasudev Alwala, Armand Joulin, and Ishan Misra. Imagebind: One embedding space to bind them all. In Proceedings of the IEEE/CVF Conference on Computer Vision and Pattern Recognition, pp. 15180–15190, 2023.
>
> > Besides TacQuad, some of the other datasets also contain tactile videos, such as TAG and OF Real. I wonder if they are already treated as videos during the training of UltraTouch, if not, maybe it can further improve the model performance if they are used as videos.
>
> Thank you for raising this point. **In fact, except for the Feel, OBF 2.0, and SSVTP datasets, which do not contain continuous video frames, all other datasets include videos, and we used these tactile videos during training**. The statistics of the training datasets we used are detailed in Table 5 in the appendix. The ablation experiments in Table 6 also show that training the model with tactile videos for dynamic perception can benefit the model's static perception abilities as well. This further supports the motivation behind our proposed method of learning unified multi-sensor representations from both static and dynamic perspectives. We have clarified this point in the revised version. Thank you again for your comment!

---

> > ### Author Response · Authors · 2024-11-24
> >
> > > The paper proposes a novel calibration platform for collecting fine-grained spatio-temporal aligned data, and they claim that this can be used for fine-grained tasks such as cross-sensor generation. However, it seems that they don't explore much on the potential of this data, i.e., most of the downstream tasks are performed on static perception, and the only pouring task is not closely related to this part of data. It would be interesting to see if any more impressive tasks can be performed using the proposed dataset.
> >
> > Thank you for your constructive comments. To more comprehensively demonstrate the value and impact of the dataset we proposed, we conducted cross-sensor generation experiments on the fine-grained spatio-temporal aligned data. Specifically, we trained models to generate aligned DuraGel images from GelSight Mini images, and to reconstruct the 20x20 force fields captured by the Tac3D sensor from DIGIT and GelSight Mini data. We compared the performance of our model with the T3 model, which used more training data than ours (3.08M compared to our 2.48M) for pretraining. Specifically, for generating Duragel images, we constructed a GAN network based on ViT, using T3 or UltraTouch as the encoders for the discriminator and generator, similar to ViTGAN [3]. A decoder is then used to generate images across sensors. For the force field generation of Tac3D, due to its low resolution, we treat it as a regression task and use an MLP to reconstruct the force field based on the features extracted by the encoder. Both networks can effectively evaluate the quality of the encoder's tactile representations. To further ensure fairness, we also removed the overlapping portions of the coarse-grained aligned data from the training data that overlapped with this dataset. **Note that Tac3D is an unseen sensor for both of the models**. We use mean-square error (MSE) (↓) between the generated data and the ground truth as the metric. The results are as follows:
> >
> > Table I: Performance comparison with T3 on the cross-sensor generation task using the fine-grained spatio-temporal aligned data (MSE as metric).
> >
> > | Method     | Training Data | GelSight Mini -> DuraGel | GelSight Mini -> Tac3D | DIGIT -> Tac3D |
> > | ---------- | ------------- | ------------------------ | ---------------------- | -------------- |
> > | T3         | 3.08M         | 0.2261                   | 0.0167                 | 0.0155         |
> > | UltraTouch | 2.48M         | **0.2159**               | **0.0151**             | **0.0144**     |
> >
> > The results show that our method outperforms T3 in terms of generation quality, both for cross-sensor generation of vision-tactile images and for force fields captured by the unseen Tac3D. This demonstrates the effectiveness of our method and the value of the dataset. This supports our motivation to obtain a unified tactile multi-sensor representation that is applicable to a variety of tasks and sensors. We also look forward to further expanding the scale, diversity, and tasks of the dataset in future work, such as tactile videos understanding and question answering, which require dynamic tactile perception. We have added the experimental results and related discussions to the revised version. Thank you again for your constructive suggestions!
> >
> >
> > [3] Kwonjoon Lee, Huiwen Chang, Lu Jiang, Han Zhang, Zhuowen Tu, and Ce Liu. Vitgan: Training gans with vision transformers. In International Conference on Learning Representations, 2022.

---

> > > ### Author Response · Authors · 2024-11-24
> > >
> > > > I'm curious about the scalability of the calibration platform used for data collection. Is it easy for other researchers to build a similar platform to scale up the current dataset?
> > >
> > > Thank you for your valuable question. The calibration platform we built consists of three main parts: a platform, a movable end effector, and a 3D-printed container that holds the sensor. These three items can be easily obtained or replaced in the laboratory. Any end effector with a coordinate system that can move to a specified position at a consistent speed (e.g., a robotic arm) is suitable for collecting fine-grained spatio-temporal aligned data. The challenge lies in how to mount multiple sensors on the end effector simultaneously while minimizing the risk of collision. A simple approach is to mount two different sensors on the finger tips of the gripper. Custom 3D-printed components can also be used to simultaneously hold multiple sensors. We would be glad to share our design files as an example.
> > >
> > > In addition to the fine-grained alignment data collection method using the calibration platform, the coarse-grained spatial aligned data collection method we proposed also holds significant potential for scaling up. In our data collection process, two skilled operators working together with four sensors to capture data from the same object can complete the task in approximately two minutes. By adding more personnel, a large volume of coarse-grained aligned multi-sensor data can be gathered in a short amount of time. We also look forward to further expanding the scale, diversity, and variety of tasks of the dataset in future work. Thank you again for your valuable question!

---

> > > > ### Comment · Reviewer_TLvC · 2024-11-26
> > > >
> > > > Thanks for the detailed reply! My major concerns are addressed, so I'll keep my rating and vote towards acceptance.
> > > > I would suggest the authors to add the additional experiments into the revised version.

---

> > > > > ### Author Response · Authors · 2024-11-27
> > > > >
> > > > > Thank you for recognizing the contribution of our work! We have added the additional experiments to the appendix of the revised version. Your constructive comments and suggestions have been invaluable to us.

---

### Official Review · Reviewer_DUex · 2024-11-04

**Soundness:** 3
**Presentation:** 3
**Contribution:** 2
**Rating:** 6
**Confidence:** 4

**Summary:**

The author introduces an aligned multi-sensor tactile dataset to unify tactile data. The data across four optical-based sensors. Also, the author proposes a unified static-dynamic multi-sensor representation learning. The author also implements real world experiments to test their framework.

**Strengths:**

(1) The idea of aligned multi-sensor dataset, improve generalization across different sensors.
(2) Capturing both static and dynamic tactile features. This is novel to make the system have comprehensive tactile information understanding.
(3) The dataset labeling process involves language model. This could improve the efficiency of the dataset collection.
(4) Using t-SNE visualization is clear to show the impact of components on representation space.
(5) I recognize the authors' effort to collect large volume dataset of tactile

**Weaknesses:**

Major Weaknesses:
1. Although the author's idea of the aligned multi-sensor dataset, the author focuses to optical-based tactile sensors only, which simplifies the problem with similar tactile modalities(tactile image).
2. In sec 5.2, the conclusion of the material classification task seems contradict to the motivation to build the aligned multi-sensor dataset.
3. The experiment results are not convincing in Section 5.5.
4. Some components of methods are not proved effective in the experiment.
4. No limitations and future work are described in the main paper.

Minor Weaknesses:

1. The presentation needs to be improved. Readers may get confused for some point before reading the supplementary material

2. The author use too many colors to represent different sensor types in the main text, which may not be good for reading.

3. Some conclusions according to experiment results are not reasonable.

**Questions:**

1. For those four types of sensors, how many sensors you chose for each type? This is important because the tactile image can be very different even the same types of sensor. The dataset could be more diverse if more sensors for each type are used.

2. [line 189]. The author mentions that collecting fine-grained aligned tactile data is very costly. Please specify the reasons.

3. [line 431]. Incorporating data from more sensors leads to a performance drop. This may potentially reflect the robustness of the mode is not good enough. Using a single sensor for each type may hard to extract effective features of this type of sensor and only focus on some specific details of a single sensor.

4.[line 488-498] The performance of UniTouch and UltraTouch is similar, it is hard to claim the advantages of the current method.

5. Please mention real-world tasks in more detailed in the main paper. The reader will be confused about the task and the meaning of "mean error" before reading the appendix.

6. It would be better if the author have experiments to prove the effectiveness of the text in this dataset.

7. [line 450-453] Please explain more details for this.

---

> ### Author Response · Authors · 2024-11-24
>
> We would like to express our sincere appreciation for your insightful comments. We appreciate your recognition of the novelty of our proposed method combining static and dynamic approaches, as well as the experimental setup and the effort put into collecting the tactile datasets.
>
>
>
> > Although the author's idea of the aligned multi-sensor dataset, the author focuses to optical-based tactile sensors only, which simplifies the problem with similar tactile modalities(tactile image).
>
> Thank you for your insightful comment. We agree that vision-based tactile sensors are just one branch of tactile sensors, and many other types of tactile sensors are also being used. However, we would like to emphasize that vision-based tactile sensors we focus on are a widely studied and utilized category of tactile sensors, and have recently received substantial attention in both academia [1] and industry [2]. Our motivation is that even within vision-based tactile sensors, there is considerable variety and a low level of standardization. As a result, different vision-based tactile sensors may exhibit differences when perceiving the same tactile information. Consequently, different vision-based tactile sensors may exhibit variations when perceiving the same tactile information. To obtain tactile representations adaptable to a wide array of tasks and sensors, we propose to learn unified multi-sensor representations from a fresh perspective that incorporates both static and dynamic perception. We think that this is already a daunting and meaningful task.
>
> We also believe that other similar types of tactile sensors can share a unified multi-sensor representation space, such as the widely used tactile sensor arrays, which can be considered a low-resolution version of vision-based tactile sensors to some extent. We look forward to further exploring other tactile sensors and their representations in future work, contributing to the continued advancement of the tactile community. We will include the related discussion in the revised version. Thank you again for the insightful comment!
>
> [1] Sudharshan Suresh, Haozhi Qi, Tingfan Wu, Taosha Fan, Luis Pineda, Mike Lambeta, Jitendra Malik, Mrinal Kalakrishnan, Roberto Calandra, Michael Kaess, Joseph Ortiz, and Mustafa Mukadam. Neuralfeels with neural fields: Visuotactile perception for in-hand manipulation. Science Robotics, 9(96):eadl0628, 2024. doi:10.1126/scirobotics.adl0628.
>
> [2] Mike Lambeta, Tingfan Wu, Ali Sengul, Victoria Rose Most, Nolan Black, Kevin Sawyer, Romeo Mercado, Haozhi Qi, Alexander Sohn, Byron Taylor, et al. Digitizing touch with an artificial multimodal fingertip. arXiv preprint arXiv:2411.02479, 2024.
>
> > No limitations and future work are described in the main paper.
>
> Thank you for your constructive suggestions! Due to space constraints, we could not fully discuss the limitations of our method earlier. Here,  we discuss some potential limitations of our work and propose corresponding solutions for future work:
>
> 1. **Compared to all the training data, the scale of the TacQuad dataset we have currently collected is still somewhat limited**. Capturing the immense variety of object types within a single dataset is challenging in a limited amount of time. Fortunately, the coarse-grained spatial alignment data collection method we propose has the potential to scale up, as data collection can be performed manually without the need for precise alignment. Fine-grained data collection can also be expanded by replicating the calibration platform and increasing manpower. We plan to grow our team to scale up the dataset and enhance object diversity in future work.
> 2. **The types of sensors considered are relatively limited**. We have made every effort to collect all available vision-based tactile sensors around us, yet we were only able to include four different types. Moreover, we did not explore the differences between individual sensors of the same type or address issues such as gel damage. Moving forward, we aim to expand our dataset and increase the diversity of sensors through collaborative data collection across multiple laboratories.
> 3. **The scope of tasks for dynamic tactile perception is currently limited**. In this work, we validated the dynamic perception capabilities of our model on a single real-world manipulation task: pouring. We hope to explore more challenging and interesting dynamic perception tasks in future work. Additionally, beyond real-world manipulation tasks, studying tactile video understanding—particularly fine-grained dynamic tactile understanding that includes direction and action descriptions—is also an interesting direction to explore.
>
> We have incorporated these discussions in the revised version. Thank you again for your valuable feedback!

---

> > ### Author Response · Authors · 2024-11-24
> >
> > > In sec 5.2, the conclusion of the material classification task seems contradict to the motivation to build the aligned multi-sensor dataset.
> >
> > > Some conclusions according to experiment results are not reasonable.
> >
> > Thank you for your valuable comments. In Section 5.2, we found that training the model using only data from the GelSight sensor achieved the best performance on the TAG material classification task. In contrast, introducing a larger dataset from other sensors actually degraded performance. The reason for this phenomenon is that the **TAG dataset is included in the pre-training data and its proportion in the full dataset changes**. Due to the scarcity of tactile datasets, existing methods [3,4,5] sometimes validate downstream datasets that are already included in the pre-trained data. In this setting, **when the downstream task's dataset has a larger proportion in the pre-training data, it is naturally more likely to perform better.** This phenomenon, noted in the CLIP paper (see Figure 17) [6], suggests that when the overlap between the downstream task dataset and the pre-training dataset is higher, the model may achieve better performance on that dataset. Therefore, integrating more data reduces the proportion of TAG data in pre-training, leading to a performance decline in material classification for the seen TAG dataset. Notably, the performance on the roughness and hardness classification tasks on TAG does not decline. This is because these binary classification tasks are much simpler, and the tactile text descriptions we generated for all datasets also include these two binary attributes, which have less impact on the data distribution.
> >
> > **We also want to emphasize that when we introduce more multi-sensor data, our method shows an overall performance improvement on unseen datasets**. The results are shown as follows:
> >
> > Table I: Performance changes on the unseen datasets when introducing more multi-sensor data.
> >
> > | Method     | Tactile Training Data                                        | Feel (Grasp)     | OF 1.0 (Material) | OF 2.0 (Material) |
> > | ---------- | ------------------------------------------------------------ | ---------------- | ----------------- | ----------------- |
> > | CLIP       | /                                                            | 72.37            | 41.00             | 73.16             |
> > | UltraTouch | TAG, VisGel, Cloth                                           | 79.12 (↑6.75)    | 46.12 (↑5.12)     | 75.10 (↑1.94)     |
> > | UltraTouch | TAG, VisGel, Cloth, OF Real                                  | 79.28 (↑0.16)    | 47.55 (↑1.43)     | 75.53 (↑0.43)     |
> > | UltraTouch | TAG, VisGel, Cloth, OF Real, TVL, SSVTP, YCB-Slide           | 79.10 (↓0.18)    | 48.00 (↑0.45)     | 75.57 (↑0.04)     |
> > | UltraTouch | TAG, VisGel, Cloth, OF Real, TVL, SSVTP, YCB-Slide, Octopi   | 79.40 (↑0.30)    | 48.75 (↑0.75)     | 75.66 (↑0.09)     |
> > | UltraTouch | TAG, VisGel, Cloth, OF Real, TVL, SSVTP, YCB-Slide, Octopi, **TacQuad** | **80.53**(↑1.13) | **49.62** (↑0.87) | **76.02**(↑0.36)  |
> >
> > Such datasets or downstream applications are actually more common and closer to real-world scenarios, as it is unlikely that we can fully encompass real-world data in the pre-training data. **This aligns with the core goal of our dataset and our method: to obtain a unified tactile multi-sensor representation that is applicable to a variety of tasks and sensors**.
> >
> > **To benefit from data in other datasets, a necessary condition is that there must be transferable overlap between these datasets**. For example, samples of the "wood" material from other datasets can assist in classifying "wood" in the downstream dataset. **However, a major issue with current tactile datasets is their limited object diversity**. For instance, the OF Real dataset, which contains up to 1,165k contact frames, only includes 100 objects from 7 different categories, and some of these categories are not even present in the TAG dataset (e.g., glass). Such datasets offer limited transferable knowledge for the seen material classification dataset. **This actually supports our goal of collecting data from a wider variety of everyday objects**. **Scaling up is not just about increasing data volume, but more importantly, improving object diversity**.
> >
> > We have expanded this discussion in the revised version. Thank you again for your valuable comments!

---

> > > ### Author Response · Authors · 2024-11-24
> > >
> > > [3] Fengyu Yang, Chao Feng, Ziyang Chen, Hyoungseob Park, Daniel Wang, Yiming Dou, Ziyao Zeng, Xien Chen, Rit Gangopadhyay, Andrew Owens, et al. Binding touch to everything: Learning unified multimodal tactile representations. In Proceedings of the IEEE/CVF Conference on Computer Vision and Pattern Recognition, pp. 26340–26353, 2024.
> > >
> > > [4] Weixian Lei, Yixiao Ge, Kun Yi, Jianfeng Zhang, Difei Gao, Dylan Sun, Yuying Ge, Ying Shan, and Mike Zheng Shou. Vit-lens: Towards omni-modal representations. In Proceedings of the IEEE/CVF Conference on Computer Vision and Pattern Recognition, pp. 26647–26657, 2024.
> > >
> > > [5] Yuanhuiyi Lyu, Xu Zheng, Dahun Kim, and Lin Wang. Omnibind: Teach to build unequal-scale modality interaction for omni-bind of all. arXiv preprint arXiv:2405.16108, 2024.
> > >
> > > [6] Alec Radford, Jong Wook Kim, Chris Hallacy, Aditya Ramesh, Gabriel Goh, Sandhini Agarwal, Girish Sastry, Amanda Askell, Pamela Mishkin, Jack Clark, et al. Learning transferable visual models from natural language supervision. In International conference on machine learning, pp. 8748–8763. PMLR, 2021.
> > >
> > > > [line 488-498] The performance of UniTouch and UltraTouch is similar, it is hard to claim the advantages of the current method.
> > >
> > > > The experiment results are not convincing in Section 5.5.
> > >
> > > Thank you for your comments. The results in Table 3 clearly demonstrate the advantages of our model over UniTouch, especially on the OF 1.0 dataset. For these two datasets, due to inherent issues with the datasets themselves, the performance improvements are not as pronounced as those on TAG or Feel. Specifically, on the OF 1.0 dataset, UniTouch’s improvement over our CLIP baseline is only 0.3. Therefore, the performance improvement of our model over UniTouch in Table 3 is sufficient to highlight our advantage.
> > >
> > > To further validate our model's ability to extract sensor-agnostic features, we trained the model on our collected fine-grained spatio-temporal aligned data for cross-sensor generation. Specifically, we trained models to generate the 20x20 force fields captured by the Tac3D sensor from DIGIT and GelSight Mini data. We compared the performance of our model with the T3 model, which used more training data than ours (3.08M compared to our 2.48M) for pretraining. We treat this task as a regression task and use an MLP to reconstruct the force field based on the features extracted by the encoder. To further ensure fairness, we also removed the overlapping portions of the coarse-grained aligned data from the training data that overlapped with this dataset. **Note that Tac3D is an unseen sensor for both of the models**. We use mean square error (MSE) (↓) between the generated data and the ground truth as the metric. The results are as follows:
> > >
> > > Table II: Performance comparison with T3 on the cross-sensor generation task using the fine-grained spatio-temporal aligned data (MSE as metric).
> > >
> > > | Method     | Training Data | GelSight Mini -> Tac3D | DIGIT -> Tac3D |
> > > | ---------- | ------------- | ---------------------- | -------------- |
> > > | T3         | 3.08M         | 0.0167                 | 0.0155         |
> > > | UltraTouch | 2.48M         | **0.0151**             | **0.0144**     |
> > >
> > > The results show that our method outperforms T3 in terms of generation quality. This demonstrates the superior ability of our model to extract sensor-agnostic features. This supports our motivation to obtain a unified tactile multi-sensor representation that is applicable to a variety of tasks and sensors. We have added the experimental results and related discussions to the revised version. Thank you again for your constructive suggestions!
> > >
> > > > Some components of methods are not proved effective in the experiment.
> > >
> > > Thank you for your comment. **In fact, we have already provided a detailed analysis of how the components of UltraTouch influence the performance of our method in Table 6 of the appendix**. The results demonstrate that all components of our method consistently make positive contributions across the four downstream datasets, except Stage 1, which focuses on learning fine-grained tactile details. To explain this, we analyzed the multi-sensor representation space of the UltraTouch model trained exclusively in Stage 1, as discussed in Section 5.3.  After introducing masked modeling, the representations become more centralized within each sensor, as this method focuses on pixel-level tactile features, which are sensor-dependent. However, this is not ideal for cross-sensor generalization, as we want multi-sensor tactile representations to cluster based on the object's tactile information they represent, minimizing sensor gaps. As a result, removing Stage 1 leads to an improvement in the model's performance on the unseen ObjectFolder 2.0 dataset instead of a decline. Nevertheless, learning pixel-level features in stage 1 is still meaningful for the seen sensors. We have clarified this point in the revised version. Thank you again for raising this point!

---

> > > > ### Author Response · Authors · 2024-11-24
> > > >
> > > > > The presentation needs to be improved. Readers may get confused for some point before reading the supplementary material
> > > >
> > > > Thank you for your careful reading. We have provided a more detailed expansion of the experimental setup in the main text of the revised version.
> > > >
> > > >
> > > >
> > > > > The author use too many colors to represent different sensor types in the main text, which may not be good for reading.
> > > >
> > > > Thank you for your suggestions! We have optimized the uses of color in the revised version.
> > > >
> > > >
> > > >
> > > > > For those four types of sensors, how many sensors you chose for each type? This is important because the tactile image can be very different even the same types of sensor. The dataset could be more diverse if more sensors for each type are used.
> > > >
> > > > Thank you for your constructive suggestions. In this work, we made every effort to include four types of sensors for data collection. However, due to the limited number of each type of sensor available to us, we used only one sensor of each type for data collection. We agree that using more sensors for each type could make the dataset more diverse, but this approach would be prohibitively expensive and time-consuming.
> > > >
> > > > Fortunately, the two multi-sensor collection methods of varying granularities that we propose can be extended to any number and type of sensor. Moreover, the text description of tactile attributes can serve as a bridge to cover the discrepancies in visual images caused by different collection scenarios. As a result, our data collection method is highly scalable, enabling collaborative data collection across multiple researchers and laboratories, similar to the success seen in existing multi-robot data collection efforts [7]. We aim to further enhance the diversity and scale of the dataset in future work, as you suggested. Thank you again for your constructive suggestions!
> > > >
> > > >
> > > > [7] Abby O’Neill, Abdul Rehman, Abhiram Maddukuri, Abhishek Gupta, Abhishek Padalkar, Abraham Lee, Acorn Pooley, Agrim Gupta, Ajay Mandlekar, Ajinkya Jain, et al. Open x-embodiment: Robotic learning datasets and rt-x models: Open x-embodiment collaboration 0. In 2024 IEEE International Conference on Robotics and Automation (ICRA), pp. 6892–6903. IEEE, 2024.
> > > >
> > > > > [line 189]. The author mentions that collecting fine-grained aligned tactile data is very costly. Please specify the reasons.
> > > >
> > > > Thank you for raising this important point! For fine-grained aligned data collection, we fix the four sensors side by side in a rectangular container and connect them to the movable end of a calibration platform. The movable end can be programmed to move at a specified speed to a designated position within the coordinate system defined by the base. Therefore, as long as we pre-measure the relative positions of the centers of the four sensor surfaces within the container and compensate for the relative positions during each set of data collection, we can ensure that all four sensors make contact with the object from the same initial position and at the same speed, thereby achieving fine-grained temporal and spatial alignment.
> > > >
> > > > However, before each data collection, we need to manually move the end effector to an appropriate initial position. Additionally, due to the small spacing between sensors, when one sensor makes contact with the object, it may be obstructed by another sensor, causing the collection to fail. As a result, **each time we collect a set of usable multi-sensor aligned data, it may require multiple repetitions or measurements, which is very time-consuming**. This is also the reason why we designed a coarse-grained collection method. We will provide a more detailed explanation of this in the revised version. Thank you again for your comment!
> > > >
> > > > > [line 431]. Incorporating data from more sensors leads to a performance drop. This may potentially reflect the robustness of the mode is not good enough. Using a single sensor for each type may hard to extract effective features of this type of sensor and only focus on some specific details of a single sensor.
> > > >
> > > > Thank you for your insightful comments. We have discussed the main reason for the performance drop when introducing more datasets from other sensors in previous responses. We also agree that using multiple sensors to collect data for each sensor type could increase the diversity of the dataset, as some sensor types may also have individual variations. However, this approach would be prohibitively expensive and time-consuming, requiring a larger number of sensors and more personnel for data collection. As our data collection method is highly scalable, enabling collaborative data collection across multiple researchers and laboratories, and we aim to further enhance the diversity and scale of the dataset in future work. Thank you again for your constructive suggestions!

---

> > > > > ### Author Response · Authors · 2024-11-24
> > > > >
> > > > > > Please mention real-world tasks in more detailed in the main paper. The reader will be confused about the task and the meaning of "mean error" before reading the appendix.
> > > > >
> > > > > Thank you for your helpful advice. For the real-world pouring task, the robot arm must rely entirely on tactile feedback to pour out 60g of small beads from a cylinder that initially contains 100g of beads. The robot arm can select one of the three actions based on the real-time tactile feedback: pouring, waiting, or retracting. Cartesian space displacement commands are generated at a policy frequency of 5 Hz. The action step size is $\delta \phi=0.25\degree$. We trained the model through imitation learning and conducted 10 test runs in the real world, recording the error between the poured mass and the target mass (100g) for each test. We averaged the error across the 10 test runs to get the "mean error" and used it as a performance metric to compare the performance. We placed some experimental setups in the appendix mainly due to page limitations. We have provided a more detailed explanation in the main paper of the revised version. Thank you again for your suggestion!
> > > > >
> > > > >
> > > > >
> > > > > > It would be better if the author have experiments to prove the effectiveness of the text in this dataset.
> > > > >
> > > > > Thank you for your valuable suggestion! To validate the effectiveness of the text in the dataset we proposed, we removed the text from the dataset and retrained the model. We then tested its linear probe performance on the four downstream tasks. The results are shown as follows:
> > > > >
> > > > > Table III: Impact of the text modality in TacQuad dataset.
> > > > >
> > > > > | Model               | TAG (Material) | Feel (Grasp)  | OF 1.0 (Material) | OF 2.0 (Material) |
> > > > > | ------------------- | -------------- | ------------- | ----------------- | ----------------- |
> > > > > | UltraTouch          | 80.82          | 80.53         | 49.62             | 76.02             |
> > > > > | w/o Text in TacQuad | 80.70 (↓0.12)  | 80.19 (↓0.34) | 49.21 (↓0.41)     | 75.91 (↓0.11)     |
> > > > >
> > > > > Although the TacQuad data is relatively small compared to the total dataset, making it unlikely to significantly impact performance when modified, we observe a consistent decline in the model's performance on downstream tasks after removing the text modality. This demonstrates the important role of the text modality in our dataset as a bridge that helps reduce the gap between sensors. We have included this experiment in the ablation study section of the revised version. Thank you again for your valuable feedback!
> > > > >
> > > > > > [line 450-453] Please explain more details for this.
> > > > >
> > > > > Thank you for your thoughtful suggestions.In Table 1, we observe that incorporating data from the DIGIT sensor into the training set causes a performance drop on the GelSight sensor datasets (TAG and Feel). Interestingly, when we further include GelSight Mini data, the performance on the GelSight datasets improves. This initially seems counterintuitive, as the combined data volume from the DIGIT sensor (39k + 4.5k + 183k = 226.5k) exceeds that of the GelSight Mini (39k). We believe this is because **the images from the DIGIT sensor differ more from those of the GelSight series, whereas the differences between GelSight and GelSight Mini, being part of the same series, are much smaller**. From a visual perspective, DIGIT sensor images feature a highly vibrant background color, whereas GelSight and GelSight Mini images have more subdued backgrounds, with strong tri-colored lighting confined to areas of deformation [8]. From a hardware perspective, the DIGIT sensor is equipped with a silicone pad known for its significant thickness and hardness [9], whereas the silicone used in GelSight and GelSight Mini is notably softer, resulting in more pronounced deformations. These differences reduce the effectiveness of directly integrating DIGIT data into training. **This also highlights the importance of the cross-sensor matching approach we propose**, which explicitly learns sensor-independent tactile features. On the contrary, the efficiency of achieving a unified multi-sensor representation solely through multi-modal alignment is highly dependent on the similarity of sensors and the distribution of object categories across different sensors. We have expanded this discussion in the revised version. Thank you again for your helpful advice!
> > > > >
> > > > >
> > > > >
> > > > > [8] Wenzhen Yuan, Siyuan Dong, and Edward H Adelson. Gelsight: High-resolution robot tactile sensors for estimating geometry and force. Sensors, 17(12):2762, 2017.
> > > > >
> > > > > [9] Mike Lambeta, Po-Wei Chou, Stephen Tian, Brian Yang, Benjamin Maloon, Victoria Rose Most, Dave Stroud, Raymond Santos, Ahmad Byagowi, Gregg Kammerer, et al. Digit: A novel design for a low-cost compact high-resolution tactile sensor with application to in-hand manipulation. IEEE Robotics and Automation Letters, 5(3):3838–3845, 2020.

---

### Author Response · Authors · 2024-11-24
**Summary of responses**

We would like to express our sincere appreciation to all reviewers for their insightful and comprehensive feedback. We sincerely appreciate the comments that the framework integrating static and dynamic perception is novel and interesting (Reviewer DUex, dEU8, oW9a and TLvC), the dataset is valuable to the community (Reviewer DUex, oW9a and TLvC), the experiments are comprehensive (Reviewer oW9a and TLvC) and the paper is well-writen and well-motivated (Reviewer dEU8).

For convenience, we have summarized some key concerns and our corresponding responses:

[Performance decline on seen datasets when more data is introduced] Due to the scarcity of tactile datasets, existing methods sometimes validate on downstream datasets that are already included in the pre-training data. In this setting, when the downstream task's dataset has a larger proportion in the pre-training data, it is naturally more likely to perform better, which is aligned with experimental findings in the CLIP paper [1]. Therefore, integrating more data reduces the proportion of the downstream data in pre-training, leading to a performance decline for the seen datasets. What we want to emphasize that when we introduce more multi-sensor data, our method shows an overall performance improvement on unseen datasets. This aligns with the core goal of our dataset and our method: to obtain a unified tactile multi-sensor representation that is applicable to a variety of tasks and sensors.

[Extension of the TacQuad dataset] Due to cost and manpower limitations, the scale of the multi-sensor paired data we collected is somewhat limited compared to the full dataset. Fortunately, as our data collection method is highly scalable, enabling collaborative data collection across multiple researchers and laboratories, the diversity of objects and sensors and the scale of the dataset can be enhanced in future work.

[New downstream task] To more comprehensively demonstrate the value of the dataset we proposed and the effectiveness of UltraTouch, we have conducted cross-sensor generation experiments on the fine-grained spatio-temporal aligned data. The results show that our method outperforms the previous method. This supports our motivation to obtain a unified tactile multi-sensor representation that is applicable to a variety of tasks and sensors.


[1] Alec Radford, Jong Wook Kim, Chris Hallacy, Aditya Ramesh, Gabriel Goh, Sandhini Agarwal, Girish Sastry, Amanda Askell, Pamela Mishkin, Jack Clark, et al. Learning transferable visual models from natural language supervision. In International conference on machine learning, pp. 8748–8763. PMLR, 2021.

---

### Meta-Review · Area_Chair_T8MR · 2024-12-24

**Metareview:**

The paper introduces TacQuad, a multi-modal tactile dataset, and UltraTouch, a unified static-dynamic multi-sensor representation learning framework, to enhance tactile perception and enable effective cross-sensor transfer, achieving state-of-the-art performance in both offline datasets and real-world tasks.

All reviewers acknowledge the contributions of this work, emphasizing its (1) novelty, (2) technical advancements, including the representation learning framework and the introduced dataset, and (3) clear presentation.

The authors effectively addressed the reviewers’ comments during the Author-Reviewer Discussion phase, resulting in improved scores.

All reviewers are in unanimous agreement to accept this paper. However, the AC recommends that the authors carefully revisit both the original and post-rebuttal reviewer comments to ensure all concerns are adequately addressed in a revised version of the paper.

**Additional Comments On Reviewer Discussion:**

Since the reviewers were in unanimous agreement to accept this paper, no significant discussion took place during the Reviewer Discussion phase.

---

### Decision · Program_Chairs · 2025-01-22

Accept (Poster)